

**Atmospheric stratification over Namibia and the southeast Atlantic Ocean**
Danitza Klopper[1,2], Stuart J. Piketh[1], Roelof Burger[1], Simon Dirkse[3] and Paola Formenti[4]
[1]North-West University, School for Geo- and Spatial Sciences, Potchefstroom, South Africa
[2]University of Limpopo, Department of Geography and Environmental Studies, Polokwane, South Africa
[3]Namibia Meteorology Service, Windhoek, Namibia
[4]Université de Paris and Univ Paris Est Creteil, CNRS, LISA, F-75013 Paris, France
*Correspondence to*: Stuart John Piketh (stuart.piketh@nwu.ac.za)
**Abstract**. We currently have a limited understanding of the spatial and temporal variability in vertically stratified
atmospheric layers over Namibia and the southeast Atlantic (SEA) Ocean. Stratified layers are relevant to the
transport and dilution of local and long-range transported atmospheric constituents. This study used eleven years
of global positioning system radio occultation (GPS-RO) signal refractivity data (2007-2017) over Namibia and
the adjacent ocean surfaces, and three years of radiosonde data from Walvis Bay, Namibia, to study the character
and variability in stratified layers. From the GPS-RO data and up to a height of 10 km, we studied the spatial and
temporal variability in the point of minimum gradient in refractivity, and the temperature inversion height, depth
and strength. We also present the temporal variability of temperature inversions and the boundary layer height
(BLH) from radiosondes. The BLH was estimated by the parcel method, the top of a surface-based inversion, the
top of a stable layer identified by the bulk Richardson number ($R_N$), and the point of minimum gradient in the
refractivity (for comparison with GPS-RO data). A comparison between co-located GPS-RO to radiosonde
temperature profiles found good agreement between the two, and an average underestimation of GPS-RO to
radiosonde temperatures of -0.45 ± 1.25°C, with smaller differences further from the surface and with decreasing
atmospheric moisture content. The minimum gradient (MG) of refractivity, calculated from these two datasets
were generally in good agreement (230 ± 180 m), with an exeption of a few cases when differences exceeded 1000
m. The surface of MG across the region of interest was largely affected by macroscale circulation and changes in
atmospheric moisture and cloud, and was not consistent with BLH($R_N$). We found correlations in the character of
low-level inversions with macroscale circulation, radiation interactions with the surface, cloud cover over the
ocean and the seasonal maximum in biomass burning over southern Africa. Radiative cooling on diurnal scales
also affected elevated inversions between 2.5 and 10km, with more co-occurring inversions observed at night and
in the morning. Elevated inversions formed most frequently over the subcontinent and under subsidence by high-
pressure systems in the colder months. Despite this macroscale influence peaking in the winter, the springtime
inversions, like those at low levels, were strongest.
**Keywords.** boundary layer, inversions, Namibia, southeast Atlantic, vertical structure
**1. Introduction**
Southern Africa is positioned in the tropics and middle latitudes where atmospheric circulation is primarily affected
by the high-pressure belt under the descending limb of the Hadley cell (Tyson and Preston-Whyte, 2014). Large-
scale subsidence of air masses below 500 hPa along this high-pressure belt results in the adiabatic heating of layers
aloft, which stabilises, and often decouples the lower troposphere, resulting in the formation of stratified
atmospheric layers (Wood and Bretherton, 2004; Painemal et al., 2014; Tyson and Preston-Whyte, 2014).





In the free troposphere over the subcontinent, four stable layers with varying degrees of persistence were identified,
at 850hPa (approximately 1.5 km), 700 hPa (approximately 3.5 km), 500 hPa (approximately 4-6 km) and 300 hPa
(approximately 8-9 km) by various authors (e.g., Preston-Whyte et al., 1977; Cosijn and Tyson, 1996; Swap et al.,
1996). The 850 hPa stability structure only formed below the elevation of the continental plateau up to the great
escarpment (up to 1000 m above sea level). These studies made use of traditional radiosondes released daily or
more frequently during short-term intensive field campaigns, resulting in a lack of spatial and temporal data
coverage.
Temperature is an important indicator of stability. Atmospheric circulation in the southeast Atlantic (SEA)
anticyclone as well as year-round (wintertime minimum) cold sea surfaces caused byupwelling in the Benguela
Current along the Namibian coast, contribute to the stabilisation of the atmospheric boundary layer (BL) (Taljaard,
1995; Cook et al., 2004; Tyson and Preston-Whyte, 2014; Gordon et al., 2018). The subsequent stability results in
a smaller temperature lapse than the adiabatic lapse rate (Warner, 2004), and is conducive to frequent fog formation
at the surface (Seely and Henschel, 1998; Haenslaer et al., 2011). High humidity and suppressed vertical cloud
growth at the top of the BL often results in the formation of a semi-permanent stratocumulus cloud (Sc) deck along
the Namibian coast (Wood, 2015). This deck extends to geographical boundaries between 10°–30°S and 10°W–
10°E (Klein and Hartmann, 1993; Muhlbauer et al., 2014), where cloud fractions range from approximately 50 to
75% in the summer/autumn and winter/spring (Muhlbauer et al., 2014). In these stable conditions, fast-moving,
diurnally-varying jet streams such as the low-level Benguela jet, form (Rife et al., 2010; Nicholson, 2010). Mean
vertical motions within the BL controls the depth- and dilution of atmospheric constituents and their potential to
be transported into the overlying air.
Vertical mixing and dilution of atmospheric constituents above the BL is largely dependent on the location and
conditions in the stratified layers aloft, and is greatest under unstable conditions (Cosijn and Tyson, 1996; Tyson
and D`Abreton, 1998; Burger, 2016). The investigation into atmospheric stratification over southern Africa has
garnered much interest over the years. Studies by Cosijn and Tyson (1996), Garstang, et al. (1996) (for non-rain
days), Tyson et al. (1996), Freiman and Piketh (2003) and the Southern African Fire Atmosphere Research
Initiative (SAFARI 2000, Swap et al., 2003) field campaign, have shown how the large-scale transport and
recirculation of aerosols from industrial pollution in the south African Highveld and biomass burning fires over
the southern African subcontinent occurs in stable layers and is transported off the west coast.
In this paper, we present the first observational study of atmospheric discontinuities over Namibia and the SEA
based on global positioning system radio occultation (GPS-RO) data from the Constellation Observing System for
Meteorology, Ionosphere, and Climate (COSMIC) spaceborne mission data collected between 2007 and 2017.
These observations provide a statistical view of the spatial and temporal variability of atmospheric discontinuities,
such as temperature inversions. As a complement, this paper also presents the results of the diurnal and seasonal
variability of the boundary layer height (BLH) obtained by the analysis of 3 years of radiosonde data launched by
the Namibia Meteorology Service from Walvis Bay airport (22°58'43.5"S 14°38'28.4"E, 97 m above mean sea
level), within the coastal margin of Namibia.
**2.    Region of investigation**
The area of interest for this study, shown in Figure 1, extends between 15°–30°S and 0°–20°E. It was divided into
three regions, namely, *coastal margin* (outlined in blue in Figure 1), *ocean* (to the west), and the *subcontinent* (to





the east), separated base on *a priori* information about variability in atmospheric stratification over similar coastal
regions.

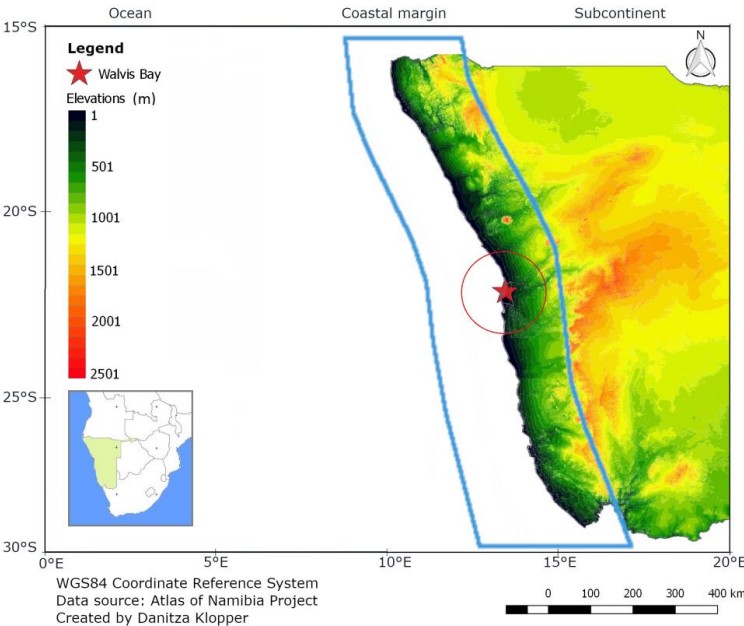

**Figure 1: Relief map of Namibia over the greater area of interest. Walvis Bay is indicated by the red star and is circled by a 100 km radius. This greater area was divided into three smaller regions, namely, *ocean* (to the west), *coastal margin* (outlined in blue), and the *subcontinent* (to the east).**

**3.    Dataset**
**3.1. COSMIC GPS-RO**
Limb-soundings of the atmosphere were made by the low earth-orbiting Global Positioning System (GPS) of the
COSMIC satellite mission that has been operational since 2006 (Anthes et al., 2008; Rocken et al., 2000). These
data cover the globe and made measurements over the region of interest several times a day (Guo et al., 2011;
Hande, 2015). The raw data were processed by the COSMIC Data Analysis and Archive Centre (CDAAC)
inversion software, utilising a Radio Occultation (RO) technique. Data are provided in the "dry atmospheric
retrieval profiles", or "atmPrf" dataset, with a vertical resolution of approximately 50–100 m in the troposphere
(Guo et al., 2011; Hande, 2015). The horizontal resolution in the troposphere is between 160 and 200 km, due to
the occultation angle (Sun et al., 2013). In coastal regions where elevation is highly variable, the coarse horizontal
resolution introduces some uncertainty in estimations of BLH.
The Abel inversion algorithm was applied to retrieve refractivity ($N$) from bending angle profiles by wave optics,
to perform quality control, as well as correct the observations for systematic negative refractivity biases and
multipath propagation of the signal (Ao et al., 2003; Sokolovskiy, 2003; Xie et al., 2006). The optimised
refractivity profile is related to temperature ($T$, K), pressure ($P$, hPa) and vapour pressure ($Pw$, hPa) of the layer
according to Equation 1 (Bean and Dutton, 1968; Guo et al., 2011).
$$N = 77.6\frac{P}{T} + 3.77 * 10^5 \frac{Pw}{T^2} \qquad\qquad (1)$$



To correct the sensitivity of the signal to moisture (Kuo et al., 2004; Seidel, 2010; Hande et al., 2015; Ho et al.,
2015), a "1-D var" moisture correction from the European Centre for Medium-Range Weather Forecasts
(ECMWF) is applied to the "atmPrf" dataset to produce the "wetPrf" dataset used in this study (O`Sullivan et al.,
2000; Shyam, 2019). In the tropical, lower troposphere, the "wetPrf" dataset has a vertical resolution of
approximately 100 m (Guo et al., 2011; Hande, 2015). It is important to note that even with the applied corrections,
no reliable information about the atmospheric structure can be collected below the point where the refractivity
signal is super-refracted (Sokolovskiy, 2003). The uncertainties due to high humidity, are related to the sensitivity
of GPS-RO signal refraction to the partial pressure of water vapour (Seidel et al., 2010; Hande et al., 2015). Chen
et al. (2011) quantified the seasonal, vertical and latitudinal dependence of observational errors of the COSMIC
GPS-RO profiles. They found that errors decreased with height in the troposphere and reported larger errors in
summer due to added moisture in the lower troposphere. Consequently, in winter the errors are more latitude
dependant than in the summer due to greater zonal moisture gradients.
Vertical refractivity profiles for 2007 to 2017 extending between 15°–30°S and 0° to 20°E were used to analyse
the long-term temporal and spatial variability in stable discontinuities near the surface and up to 10 km. The
frequency of measurements is summarised in Figure S.1, showing the uneven spatial and temporal distribution
across the three regions identified in Figure 1. Months with less than 10 measurements per region were excluded
from the analyses. The number of valid profiles available for each region was 18475 (ocean), 4007 (coast) and
4369 (subcontinent).

### 3.2. Radiosondes

Radiosonde data were measured at Walvis Bay airport by the Namibian Meteorology Service using an iMet-2-AA
radiosonde instrument, which was operated according to the latest World Meteorological Organization's (WMO)
*Guide to Instruments and Methods of Observation* (2017). Standard variables contained in the dataset are
temperature, pressure, wind speed and -direction, relative humidity, and elevation with 50-m vertical resolution.
Measurements were interactively quality controlled by automated statistical regression to remove any erroneous
values (WMO, 2017). To standardise the time of measurement, local Namibian time of radiosonde release was
converted to UTC, as UTC +1 for measurements between the first Sunday in April to the first Sunday in September,
otherwise UTC +2. A total of 335 soundings for 2015 are retained for the analyses of the annual variability, as it
was the most complete year of measurements. These radiosondes were generally launched at 10h00 local time,
with some launches at 09h00.  We also include 232 radiosondes from 2014 and 2016, released between 10h00 and
11h00 local time, for additional comparisons of estimated boundary layer heights and temperature profiles with
GPS-RO data.

### 4.  Data analysis

#### 4.1. Point of minimum gradient in refractivity and vapour pressure

The bending angles of GPS-RO refractivity profiles are proportional to atmospheric moisture. As a consequence,
assuming that the top of the BL marks a sharp transition from the free troposphere to a more dense, moist, and
refractive layer, the BLH was estimated as the height of the point of the relative minimum gradient (MG) and $\boldsymbol{Pw}$
in the refractivity profile (Xie et al., 2006; Sokolovskiy et al., 2007; Guo et al., 2011; Ao et al., 2012; Ho et al.,
2015). This definition assumes a neutral atmosphere, as given in Equation 1.





In a global climatological study, Ao et al., (2012) investigated seasonal BLHs over the ocean in our region of
interest, using GPS-RO and radiosonde data, and found that BLHs from GPS-RO were on average 100 m higher
in the autumn and 100 m lower in the spring. Ao et al. (2012) also found that GPS-RO showed greater variability
than era interim data, especially in the tropics. Ho et al. (2015) showed how the MG method was useful over
marine areas in the southeast Pacific off South America, where they considered the thick, semi-permanent Sc
(Warner, 2004) which tops the marine BL at ~850 hPa over the SEA (Muhlbauer et al., 2014; Wood, 2015) as the
BLH. This region shares similar climatic conditions to the coastal desert environment of Namibia. The point of
MG of refractivity (hereafter MG height) and water vapour profiles were therefore considered potential definitions
of BLH and this definition was tested in section 5. We set a maximum height of 4000 m in which to find the MG
height.
We applied smoothing over a 250 m window for the refractivity profiles to get rid of sharp changes in the profile
linked to microscale variability (Ao et al., 2003). For the comparison with radiosonde data, the GPS-RO profiles
were limited to within a radius of 100 km around the radiosonde launch point (Figure 1), which is characterised
by similar surface cover over land and includes a small portion of ocean.
**4.2.  Calculation of boundary layer height from radiosonde data**
The comparative analysis of the radiosonde profiles was performed based on four different definitions of BLH, to
evaluate the sensitivity of the results to the definition used. First, in analogy with the GPS-RO treatment, the
radiosonde profiles of pressure, temperature, and moisture were converted into a refractivity profile according to
Equation 1 (Xie et al., 2012; Hande et al., 2015) and subsequently used to estimate BLH with the MG method.
Additionally, the BLH from radiosonde observations was defined as the top of a surface-based inversion, which
would stabilise surface layers and inhibit vertical mixing.
Boundary layer height was also defined as the point where the virtual potential temperature (VPT) aloft is the same
as at the surface, which is the height at which a parcel would be in equilibrium with its surroundings (the parcel
method; Seibert et al., 2000; Seidel et al., 2010). Lastly, the BLH was defined as the top of the stable layer,
assuming the stabilisation of the lower atmosphere along the coast by the year-round cold waters of the Benguela
current. To find this height, we used the bulk Richardson number ($R_N$) defined as;
$$R_N(h) = \frac{g(h-h_0)}{\theta(h)} \frac{[\theta(h)-\theta(h_0)]}{u(h)^2+v(h)^2} \qquad (2)$$
where $\boldsymbol{\theta}$ is potential temperature, $\boldsymbol{g}$ is gravitational acceleration, $\boldsymbol{u}$ (zonal) and $\boldsymbol{v}$ (meridional) denote wind
components at the surface ($\boldsymbol{h_o}$) and at given altitude ($\boldsymbol{h}$). A critical bulk Richardson number ($\boldsymbol{R_N}$) at height $\boldsymbol{h}$ greater
than zero identifies the height above the statically stable flow in the BL (Sørenson et al., 1996; Korhonnen et al.,
2014). An upper limit of 4000 m above ground level (agl) is set in which to find the BLH by the three above-
mentioned definitions.
**4.3.  Subsidence inversions**
Preston-Whyte et al. (1977) reported a minimum height of 0.5 km above mean sea level (amsl) for the subsidence
inversion over the Benguela Current. To exclude surface inversions over the ocean and varying topography over
the subcontinent, temperature profiles from the radiosondes were limited to between 0.5 and 10 km agl. To do the
same for GPS-RO data, profiles needed to be converted from height amsl to height above ground level (agl) by
extracting the elevation for each measured coordinate point from the Shuttle Radar Topography Missions, 30 m



raster data elevation model (SRTM, 2013). To compare heights from the two datasets and across varying
topography, the height amsl of GPS-RO data was retained, and the radiosonde measurements in height agl were
converted into height amsl, considering the 91 m surface altitude at Walvis Bay airport.
Temperature profiles were derived from the refractivity profiles using Equation 1. The fact that these temperatures
are derived and not measured directly, introduces uncertainties that are largely dependent on the moisture content
of the atmosphere and the algorithm applied or retrieval method used (Wang et al., 2013). Equation 1 was also
used to calculate the saturation mixing ratio, mixing ratio, and dew point temperatures.
Thermal inversions are defined by an increase in temperature with height. Their strength, describing the local
stability of the atmosphere, was calculated as the change in temperature per 100 m interval. Inversion depth is
simply the vertical height through which the inversion persists. Shallow isothermal layers directly above or below
the inversion were not included in the calculation of the inversion depth.
**5.    Results**
**5.1.  Comparisons between COSMIC GPS-RO and radiosonde data**
**5.1.1. Temperature**
To investigate how the derived temperature from GPS-RO profiles compare to measured temperature profiles from
radiosondes, we show all profiles in Figure 2 that were measured on the same day and co-located within 100 km
of Walvis Bay (shown in Figure 1). The comparison is particularly important at low altitudes where the GPS-RO
retrieval may be affected by moisture and clouds, and some information may be lost that is otherwise captured by
the radiosonde measurements. This is evident in the inability of any of the GPS-RO profiles in Figure 2 to capture
the surface inversions that were seen in the radiosonde profiles. In fact, none of these co-located profiles extended
below 500m. Furthermore, it's important to note the higher vertical resolution of measurements made by
radiosonde as compared to GPS-RO signals, discussed in section 3. Despite the differences in time and resolution
of measurements, the temperature profiles were in good agreement in terms of temperature range and lapse rates.
Alexander et al. (2014) reported a relative error in the refractivity profile of -0.2 % below 8km and Wang et al.
(2013), an error of ± 1.6% (3% at 1000 hPa and 0.5% at 300h Pa). The greatest biases were reported for the tropics
under high humidity conditions (Sokolovskiy, 2003; Kuo et al., 2004). Wang et al., (2013) attributed errors between
2 and 10 km to the effects of signal propagation through a dry atmosphere, and an opposite bias for errors below
2 km due to very low $Pw$ conditions (moist atmosphere) (Ao et al., 2003). For all the comparative profiles in Figure
2, the absolute error (and standard deviation) in temperatures was -0.30 ± 1.30°C below 7 km amsl for where GPS-
RO temperatures were lower than radiosondes. In the drier portion of the atmosphere, between 7 and 10 km,
comparisons in Figure 2 show an error of -0.45 ± 1.25°C. The difference measured by the two methods compared
well with the -0.20 ± 1.50°C reported by Wang et al. (2013) for global comparisons using the "wetPrf" data and
several radiosonde types. The most extreme example of the differences in detected temperature with height
between the two methods, is on 2016/07/24 (Figure 2), where the two profiles crossed around 4.5 km.

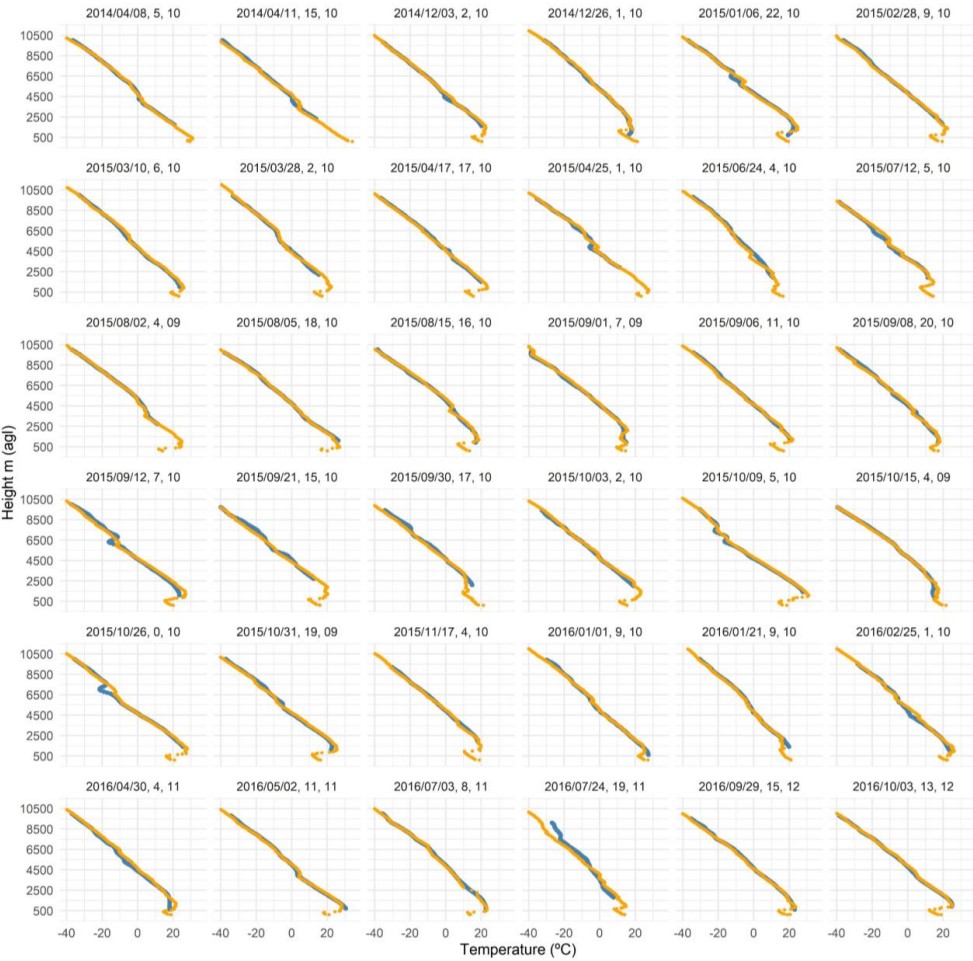

**Figure 2: Temperature profiles from GPS-RO (blue) and radiosonde (orange), taken on the same day and within 100 km from Walvis Bay. Each set of profiles is labelled with the date of measurement, hour of GPS-RO measurement and radiosonde launch time.**

**5.1.2. Minimum gradient of refractivity**
A direct comparison is made between the MG height from GPS-RO and the radiosonde data. Figure 3 presents a
scatterplot of MG heights from radiosondes and co-located GPS-RO profiles taken within six hours and a 100 km
radius from Walvis Bay. The MG height estimated from the two datasets show poor agreement with overall mean
differences in the estimated height of $790 \pm 990$ m. The MG height estimated by GPS-RO generally overestimates
the values obtained from the radiosonde data, with only a few exceptions. Biases and observational errors in GPS-
RO profiles (Chen et al., 2011) and radiosondes (Wang and Zhang, 2008; Wang et al., 2013), and measurement
resolution, have been the main reasons for a statistical difference in the retrieval of vertical atmospheric structures.
The differences in Figure 3 could not be explained by differences in time of measurement, which would affect




humidity. The location of each of these GPS-RO measurements relative to Walvis Bay is given in Figure S.2, and
shows that these differences are also not a direct result of spatial variability in the height of atmospheric layers.

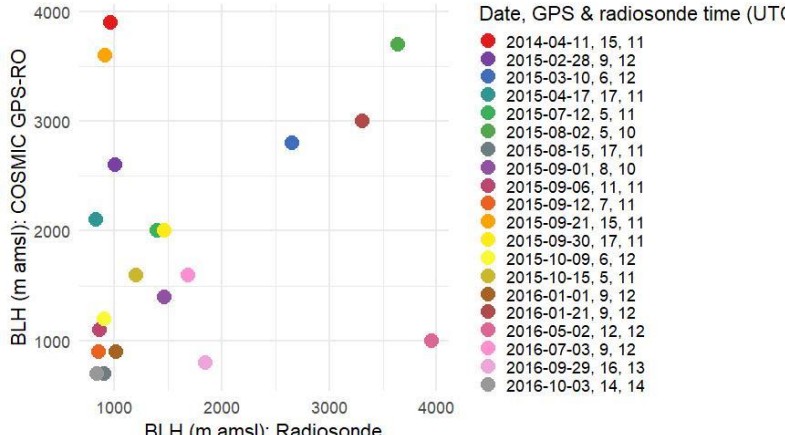

**Figure 3: Scatterplot of BLHs estimated as the MG in refractivity, identified in radiosondes and co-located GPS-RO profiles within 100 km and 6 hours of the radiosonde release. These points are labelled by date, and the hour of GPS-RO and radiosonde measurement.**

Atmospheric moisture poses the greatest challenge for both GPS-RO and radiosonde sensors, especially in the
presence of clouds (Wang and Zhang, 2008; Wang et al., 2013). The biggest differences in MG heights above
1000m, were measured for all reported cases in February to May, and for two cases in September (Figure 3).
Considering the MG heights found, it is unlikely that the differences are explained by the inability of GPS-RO
signals to penetrate to the surface beyond a point of superrefraction, such as in the presence of high cloud fractions
in low-level clouds (Ao et al., 2012). When we exclude the six points that show differences greater than 1000 m
in the estimated point of MG, we find much better agreement between the GPS-RO and radiosonde datasets, with
mean differences of 230 ± 180 m. Although we have no explanation for the largest discrepancies, the good
agreement between the remaining points shows that the layer described by the MG height is likely a real
atmospheric discontinuity, and may be relevant to air mass transport and vertical distribution of atmospheric
constituents. This discontinuity is therefore treated separately from BLH as defined in the radiosonde data.
**5.2. Spatial and temporal variability in low-level discontinuities**
**5.2.1. Boundary layer height**
This section presents the morning BLH, estimated by the $R_N$, VPT and surface-based inversion definitions, at
Walvis Bay. These measurements were made at 9 and 10 UTC in 2015, which was the most complete year-long
measurement record. The monthly mean and standard deviation in BLH for 2015 is summarised in Figure 4, and
Table S.1.

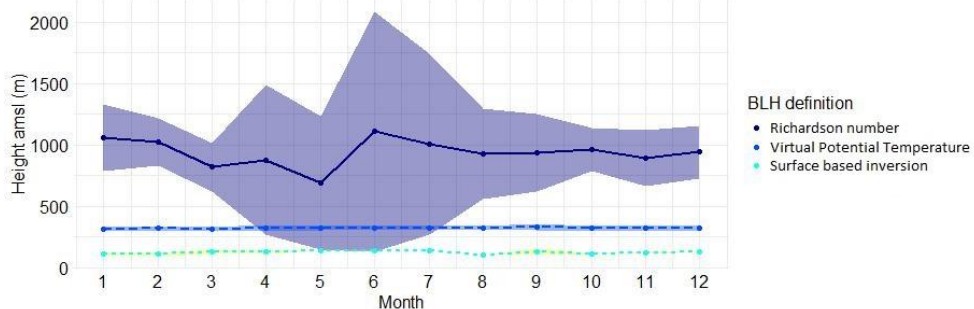

**Figure 4: Time series for 2015 of the monthly mean and standard deviation of mid-morning BLH at Walvis Bay,**
**calculated by the Bulk Richardson number, the point where virtual potential temperature above the surface is the**
**same as at the surface, and the top of a surface-based inversion from radiosonde data.**
The BLH, defined as the point where VPT is equivalent to that at the surface, remained stable around $320 \pm 10$ m,
as did the top of the surface-based inversion ($125 \pm 10$ m) (Figure 4 and Table S.1). The bulk $R_N$ definition found
the monthly mean minimums in the austral autumn ($809 \pm 477$ m) and spring ($934 \pm 250$ m), and maximums in
the summer ($1007 \pm 60$ m) and winter ($1018 \pm 741$ m) of 2015 (Figure 4). The range of BLHs estimated by the
bulk $R_N$ definition in Figure 4 was similar to values and seasonal trends for the region derived by Von Engeln and
Teixeira (2013) by the MG of relative humidity. We found the highest BLHs (with high variability) in the winter.
These high BLHs coincided with a distinct decrease in the height of easterly wind components, shown in Figure
5a. The overall variability in BLH decreased with a decreasing frequency of easterly wind components below 2500
m, as well as a decrease in temperature near the surface and lower temperature inversion heights, as seen in Figure
5b.

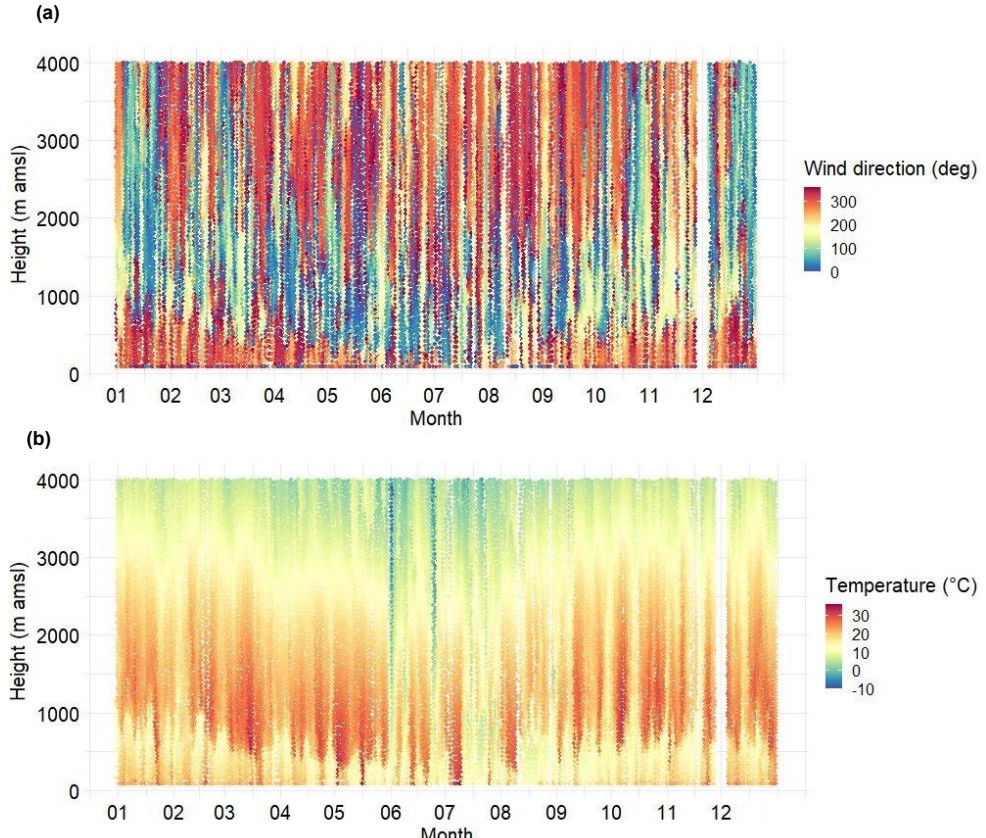

Figure 5: (a) Wind direction and (b) temperature profiles measured by radiosondes released from Walvis Bay in 2015.

### 5.2.2. Minimum gradient of refractivity

The MG height from GPS-RO and radiosondes (profiles limited to a minimum of 800 m, as discussed in section 5.2) is given in Figure 6 and summarised in Table S.2. To get the most sensible comparison between the datasets, and include the greatest number of profiles for the comparison, the GPS-RO profiles were limited to morning measurements between 7 and 12 UTC and within 100 km of Walvis Bay. The inter-annual variability is in rather good agreement, however, the mean difference between MG height estimated from GPS-RO and radiosonde is large (550 ± 290 m). The smallest differences between the datasets were measured in March, May, June, September, and October. The interannual variability and heights of the BLH ($R_N$) in Figure 4 and MG heights in Figure 6 suggests that these identified layers are not synonymous.



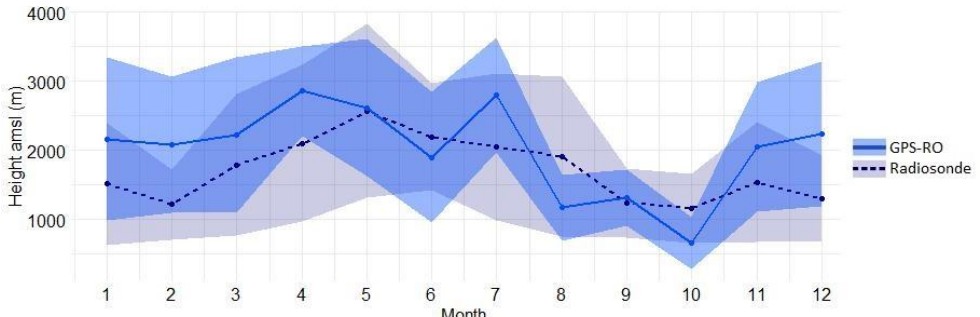

**Figure 6: The minimum gradient of refractivity calculated from GPS-RO (2007-2017) and radiosonde data (2014-**
**2016), where radiosonde profiles were limited to higher than 500 m amsl.**
The seasonal mean and standard deviations of the MG height over the region of interest is given in Figure 7
(monthly average and standard deviations are given in Figure S.3). The overall mean and spatial variability of the
MG height averaged between 2007 and 2017 (Figure S.4) is in very good agreement with that reported by Ao et
al., (2012) for 2006 to 2009. Mean MG heights were consistently higher over land (smaller standard deviations)
than over the ocean (greater standard deviations) and the resulting meridional gradient was most steep in the winter
and least steep in the summer. From these plots, we can deduce that the standard deviations are indicative of both
the temporal variance and the influence of meteorological conditions on MG height.

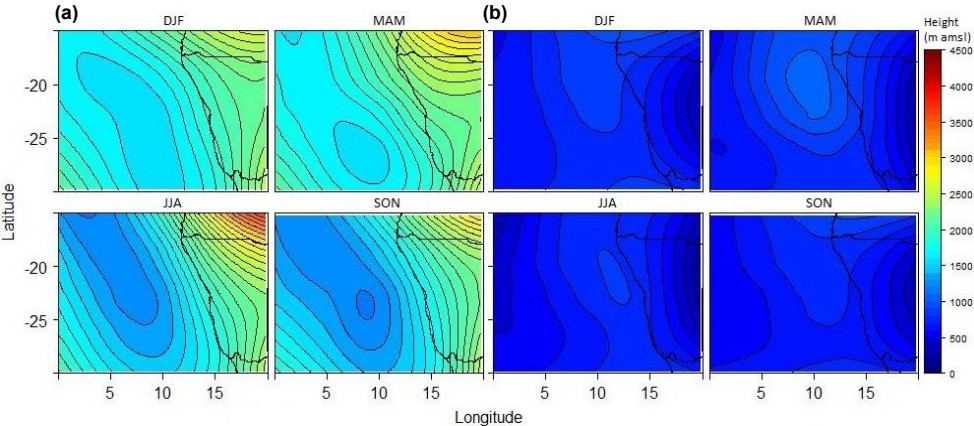

**Figure 7: The seasonal (a) mean and (b) standard deviation of the height of the MG of refractivity over the area of interest. Polynomial smoothing was applied to the data.**

The diurnal variability in mean and standard deviation of the height of MG N-refractivity profiles are given in
Figure 8a and 8b respectively. The greatest diurnal variability in BLH was over the subcontinent, with maximum
heights in the daytime (9 to 20 UTC), and minimums between 21 and 8 UTC. Over the ocean, the diurnal variability
in mean heights was smaller, although standard deviations (Figure 8b) indicate greater variance in heights within
each timeframe.

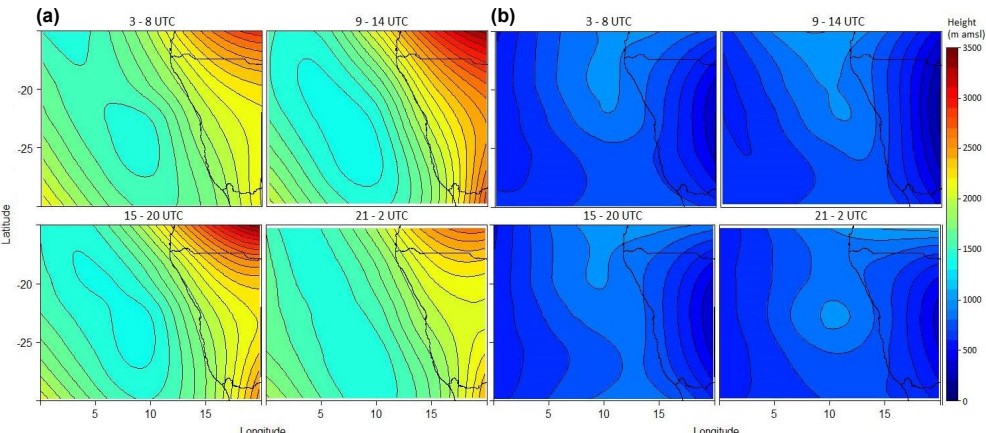

**Figure 8: The diurnal (a) mean and (b) standard deviation of the height of the MG of refractivity over the area of interest. Polynomial smoothing was applied to the data.**

**5.2.3.  Temperature inversions**
Table 1 presents seasonally averaged low-level inversion characteristics. The seasonal mean and standard deviation
of inversion base heights (between 0.5 and 2.5 km) are given in Figure 9, and the overall mean and standard
deviation are given in Figure S.5.

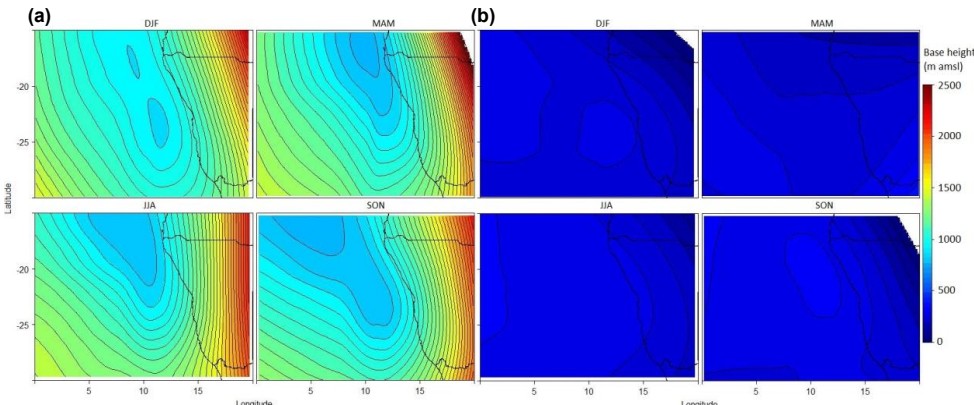

**Figure 9: The seasonal (a) mean and (b) standard deviation of the height of low-level temperature inversion over the area of interest. Polynomial smoothing was applied to the data.**

Based on equation 1, and as illustrated by Lopez, (2009), super refractivity is related to increases in temperature
with height. We found that inversion heights (Figure 9) were not concomitant with MG of refractivity heights
(Figure 7) and were on average 190 ± 480 m lower. Beyond these layers, particularly in the moist marine
atmosphere, the profile will likely be super refracted (Xie et al., 2006) and no further information may be retrieved.
We found some similarities between these two layers, particularly in regards to the orientation of height gradients.
Over the ocean, the gradient towards the lowest points in these layers were in similar locations, particularly in the





winter and early spring (Figure S.6). Between these layers, we found a correlation of 0.61, which indicates a
moderate relationship. No spatial trends in the differences in height between the two layers were found.
Low-level inversions formed most frequently over the ocean, and especially between 20° and 25°S (Table 1). The
low-level inversions over the ocean were situated around $1.1 \pm 0.3$ km amsl and over the coast around $0.9 \pm 0.3$
km, in the range of, but lower than the 850 hPa stable layer (expected around 1.4 km) described by Cosijn and
Tyson (1996). Over the ocean, Figure 9 and Table 1 shows the lowest base heights near the coast and at lower
latitudes. The zonal variability in base heights was smallest in the summer and greatest in the winter. Over the
ocean and coastal margin, and towards lower latitudes, mean depths of these inversions (Table 1) agreed with 400
– 500 m reported by Preston-Whyte et al. (1977) and 500 m reported by Cosijn and Tyson (1996) for the west
coast and Namibia.
Springtime inversions exhibited high spatial and temporal variability. Latitudinal variability in inversion strengths
was greatest in the spring, where over the ocean, inversion strengths increased towards lower latitudes (Table 1
and Figure 10) and reached an annual maximum. Additionally, north of 20°S, springtime inversions were lowest
and deepest.

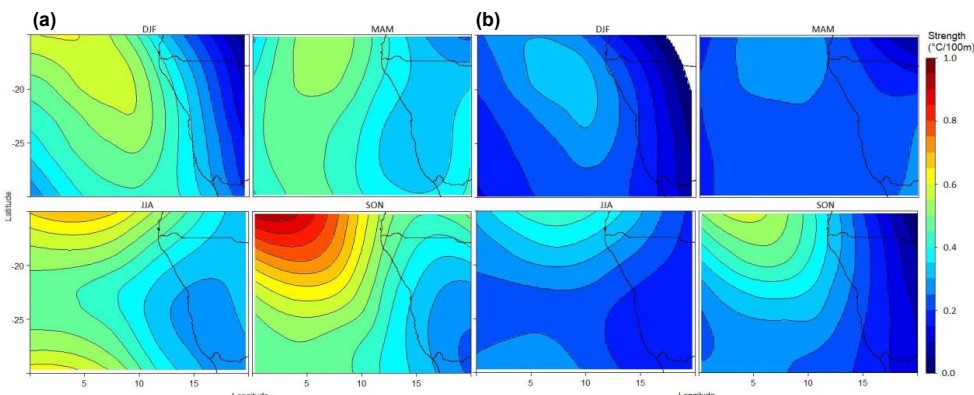

**Figure 10: The seasonal (a) mean and (b) standard deviation of the strength of low-level temperature inversion over the area of interest. Polynomial smoothing was applied to the data.**



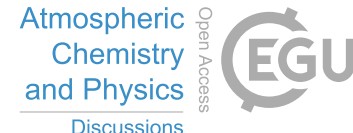


**Table 1: Seasonal mean and standard deviation of the base heights (m amsl), depths (m), strengths (°C/100 m) and strength throughout the depth (°C/depth) of low-level inversions (0.5 – 2.5 km agl) identified in the COSMIC GPS-RO data in the 15° to 20°S, 20° to 25°S and 25° to 30°S zonal bands over the region of interest, divided into oceanic, coastal and subcontinental regions. The table also shows the number of inversions measured in each region and by season (Su.=DJF, A.=MAM, W.=JJA and Sp.=SON).**

| | | 15° to 20°S | | | 20° to 25°S | | | 25° to 30°S | | |
|---|---|---|---|---|---|---|---|---|---|---|
| | | Ocean | Coast | Subcontinent | Ocean | Coast | Subcontinent | Ocean | Coast | Subcontinent |
| Height (m agl) | DJF | 950 ± 390 | 740 ± 260 | 1730 ± 350 | 1080 ± 450 | 900 ± 410 | 1950 ± 380 | 1240 ± 520 | 1310 ± 590 | 1730 ± 410 |
| | MAM | 1020 ± 330 | 870 ± 300 | 1480 ± 270 | 1040 ± 350 | 880 ± 300 | 1660 ± 180 | 1110 ± 400 | 960 ± 370 | 1420 ± 190 |
| | JJA | 1000 ± 350 | 700 ± 190 | 2430 ± 60 | 1090 ± 410 | 790 ± 270 | 1640 ± 280 | 1180 ± 460 | 980 ± 490 | 1720 ± 450 |
| | SON | 830 ± 240 | 820 ± 280 | 1530 ± 110 | 930 ± 350 | 820 ± 330 | 1650 ± 160 | 1110 ± 470 | 1080 ± 470 | 1720 ± 390 |
| Depth (m) | DJF | 400 ± 200 | 400 ± 200 | 200 ± 100 | 400 ± 200 | 400 ± 200 | 300 ± 200 | 400 ± 200 | 400 ± 200 | 300 ± 200 |
| | MAM | 400 ± 200 | 300 ± 200 | 200 ± 100 | 400 ± 200 | 400 ± 200 | 200 ± 200 | 300 ± 200 | 300 ± 200 | 200 ± 200 |
| | JJA | 300 ± 200 | 300 ± 200 | 400 ± 100 | 300 ± 200 | 300 ± 200 | 200 ± 200 | 300 ± 200 | 300 ± 200 | 400 ± 800 |
| | SON | 400 ± 200 | 500 ± 300 | 200 ± 100 | 400 ± 200 | 500 ± 300 | 200 ± 100 | 400 ± 200 | 300 ± 200 | 300 ± 200 |
| Strength (°C/100m) | DJF | 0.57 ± 0.36 | 0.46 ± 0.27 | 0.35 ± 0.19 | 0.44 ± 0.31 | 0.32 ± 0.19 | 0.32 ± 0.19 | 0.47 ± 0.37 | 0.26 ± 0.15 | 0.32 ± 0.19 |
| | MAM | 0.55 ± 0.33 | 0.41 ± 0.25 | 0.31 ± 0.21 | 0.49 ± 0.32 | 0.37 ± 0.23 | 0.21 ± 0.08 | 0.43 ± 0.31 | 0.34 ± 0.25 | 0.26 ± 0.12 |
| | JJA | 0.49 ± 0.35 | 0.43 ± 0.3 | 0.28 ± 0.14 | 0.45 ± 0.33 | 0.3 ± 0.2 | 0.62 ± 1.25 | 0.42 ± 0.32 | 0.27 ± 0.17 | 0.26 ± 0.22 |
| | SON | 0.78 ± 0.43 | 0.51 ± 0.32 | 0.42 ± 0.34 | 0.58 ± 0.37 | 0.4 ± 0.24 | 0.32 ± 0.18 | 0.46 ± 0.32 | 0.31 ± 0.21 | 0.24 ± 0.17 |
| Strength (°C/depth) | DJF | 2.32 ± 2.49 | 1.69 ± 1.63 | 0.52 ± 0.67 | 1.98 ± 2.46 | 1.14 ± 1.27 | 0.96 ± 1.24 | 2.20 ± 3.13 | 0.92 ± 0.93 | 0.83 ± 0.90 |
| | MAM | 2.08 ± 1.9 | 1.19 ± 1.17 | 0.4 ± 0.27 | 1.95 ± 2.17 | 1.46 ± 1.37 | 0.44 ± 0.48 | 1.68 ± 2.14 | 1.09 ± 1.00 | 0.54 ± 0.54 |
| | JJA | 1.48 ± 1.77 | 1.03 ± 1.04 | 1.27 ± 0.96 | 1.61 ± 2.12 | 0.8 ± 0.89 | 0.48 ± 0.50 | 1.76 ± 2.78 | 0.72 ± 0.81 | 0.54 ± 0.51 |
| | SON | 2.98 ± 2.31 | 2.28 ± 1.88 | 0.54 ± 0.66 | 2.24 ± 1.99 | 1.77 ± 1.54 | 0.49 ± 0.49 | 1.79 ± 2.23 | 0.99 ± 0.99 | 0.86 ± 1.28 |
| Number of inversions | DJF | 1115 | 190 | 47 | 1590 | 166 | 62 | 1221 | 157 | 56 |
| | MAM | 831 | 123 | 12 | 1327 | 208 | 7 | 1085 | 162 | 16 |
| | JJA | 753 | 84 | 3 | 1155 | 104 | 17 | 994 | 128 | 20 |
| | SON | 912 | 164 | 35 | 1268 | 236 | 32 | 1065 | 158 | 31 |






The seasonal variability of low-level inversion characteristics over the greater coastal region as described by GPS-
RO data, are not representative of radiosonde measurements over Walvis Bay, as seen in Table 2. Seasonal mean
inversion strengths per 100 m for low-level temperature inversions measured by radiosonde data at Walvis Bay
(Table 1) were in the order of the 2–5°C reported for summer and 3–4.5°C reported for winter over the Benguela,
by Preston-Whyte et al., (1977).
**Table 2: Seasonal mean and standard deviation of the base heights (m amsl), depths (m), strengths (°C/100 m) and**
**strength throughout the depth (°C/depth) of low-level inversions (0.5 – 2.5 km agl) from radiosondes released at**
**Walvis Bay airport.**

|  |  | 0.5 – 2.5 km |
|---|---|---|
| Height (m agl) | Su. | 980 ± 350 |
|  | A. | 910 ± 330 |
|  | W. | 1160 ± 570 |
|  | Sp. | 1020 ± 430 |
| Depth (m) | Su. | 150 ± 120 |
|  | A. | 160 ± 120 |
|  | W. | 140 ± 110 |
|  | Sp. | 170 ± 120 |
| Strength (°C/100m) | Su. | 0.01 ± 0.01 |
|  | A. | 0.01 ± 0.01 |
|  | W. | 0.01 ± 0.01 |
|  | Sp. | 0.01 ± 0.01 |
| Strength (°C/depth) | Su. | 2.72 ± 2.59 |
|  | A. | 2.16 ± 2.09 |
|  | W. | 1.91 ± 2.23 |
|  | Sp. | 3.27 ± 3.43 |
| Number of inversions | Su. | 183 |
|  | A. | 144 |
|  | W. | 73 |
|  | Sp. | 177 |

Diurnal variability was compared between measurements made in the morning (3 to 8 UTC), noon (9 to 14 UTC),
afternoon (15 to 20 UTC), and night (21 to 2 UTC). On diurnal scales, this inversion layer formed more frequently
at night and in the early morning, as seen in Table S.2, when atmospheric stability was greatest. Diurnal variability
in base heights (Figure 11) and inversion strengths (Figure 12) was smallest along the coastal margin.

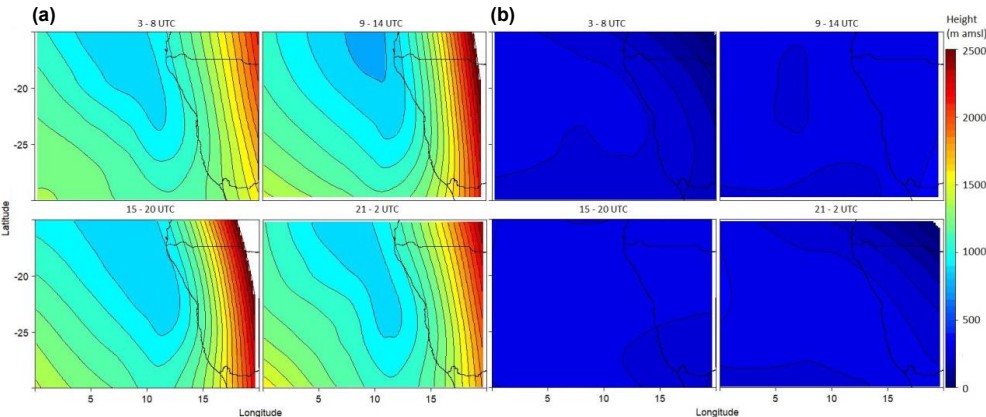

**Figure 11: The diurnal (a) mean and (b) standard deviation of low-level inversion base height over the area of**
**interest. Polynomial smoothing was applied to the data.**

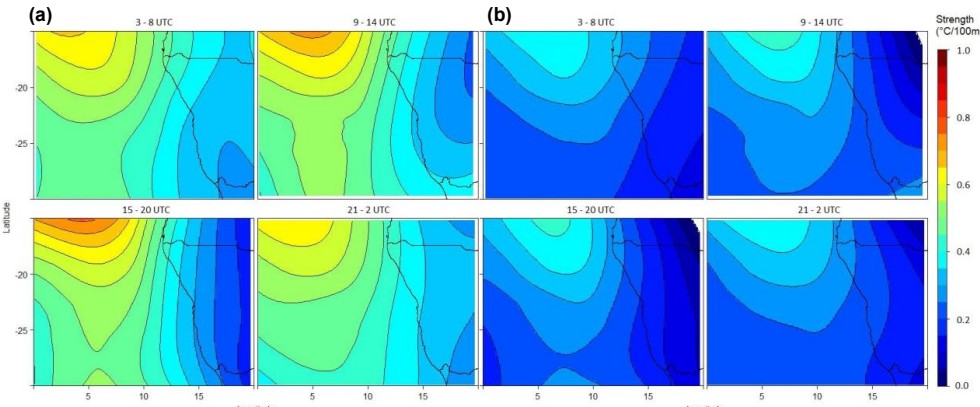

**Figure 12: The diurnal (a) mean and (b) standard deviation of low-level inversion strength over the area of interest. Polynomial smoothing was applied to the data.**

### 5.3. Spatial and temporal variability in elevated discontinuities

### 5.3.1. Temperature inversions

Table 3 presents seasonally averaged zonal summaries of elevated inversion characteristics. As with the low-level inversions, the highest frequency of elevated inversions was measured over the open ocean between 20° and 25°S. The lowest frequency of formation was along the coastal margin north of 25°S and over the subcontinent between 25° and 30°S. The overall mean (and standard deviation) of elevated inversion heights (Figure S.8) was highest aligned with and along the coast, with a southward increase in mean base heights from 20°S. Seasonal mean base heights for temperature inversions between 2.5 km and 10 km agl, given in Figure 13 (monthly means in Figure S.9), show high meridional variability year-round. Mean inversion height over the subcontinent was greatest in the spring and summer with comparatively little variability. Wintertime inversions over the subcontinent had the lowest mean heights (and increasing variability towards the south). Wintertime inversions were among the weakest, whereas springtime inversions, particularly September (Figure S.10) were strongest (Table 2 and Figure 14).




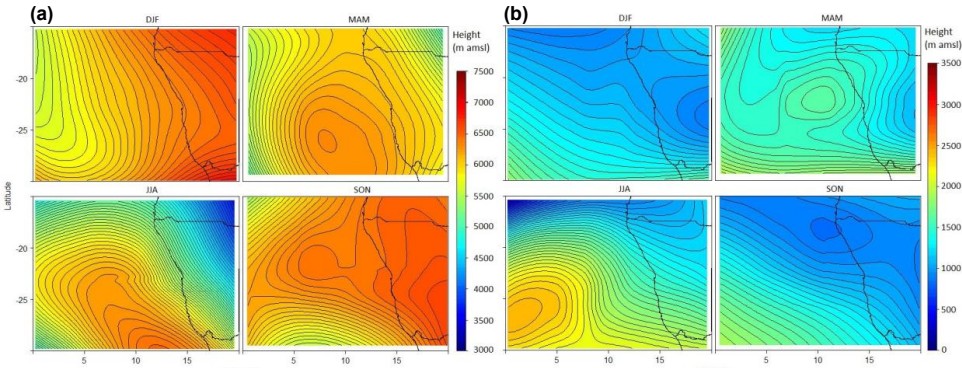

**Figure 13: The seasonal (a) mean and (b) standard deviation of the height of elevated temperature inversions over the area of interest. Polynomial smoothing was applied to the data.**

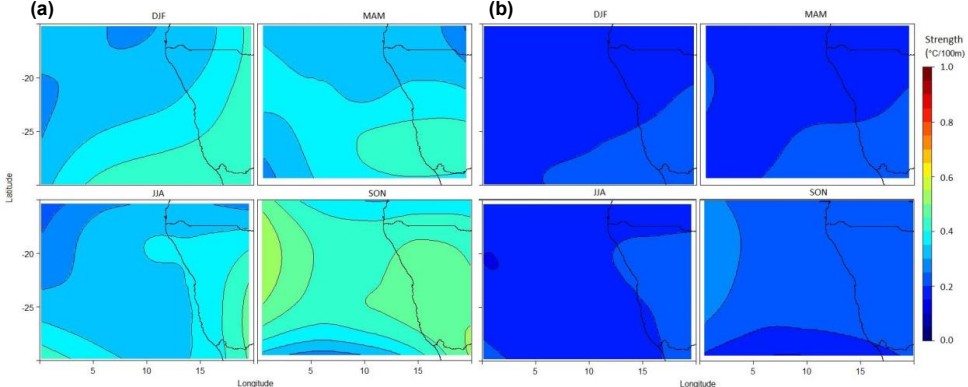

**Figure 14: The seasonal (a) mean and (b) standard deviation of the strength of elevated temperature inversions over the area of interest. Polynomial smoothing was applied to the data.**

327





**Table 3:** Seasonal mean and standard deviation of the base heights (m amsl), depths (m), strengths (°C/100 m) and strength throughout the depth (°C/depth) elevated inversions (2.5 – 10 km agl) identified in the COSMIC GPS-RO data in the 15° to 20°S, 20° to 25°S and 25° to 30°S zonal bands over the region of interest, divided into oceanic, coastal, and subcontinental regions. The table also shows the number of inversions measured in each region and by season (Su.=DJF, A.=MAM, W.=JJA and Sp.=SON).

| | | 15° to 20°S | | | 20° to 25°S | | | 25° to 30°S | | |
|---|---|---|---|---|---|---|---|---|---|---|
| | | Ocean | Coast | Subcontinent | Ocean | Coast | Subcontinent | Ocean | Coast | Subcontinent |
| Height (m agl) | DJF | 5590 ± 1980 | 5070 ± 1500 | 4230 ± 1260 | 6200 ± 2240 | 5370 ± 1810 | 4860 ± 1510 | 6320 ± 2250 | 6010 ± 2240 | 5440 ± 1790 |
| | MAM | 5960 ± 1150 | 6370 ± 1240 | 6540 ± 1160 | 6010 ± 1400 | 6290 ± 1070 | 6420 ± 1160 | 5880 ± 1410 | 6390 ± 1290 | 6570 ± 1140 |
| | JJA | 5920 ± 1540 | 5940 ± 1360 | 5710 ± 1610 | 6240 ± 1750 | 6130 ± 1430 | 6010 ± 1330 | 6180 ± 1720 | 6040 ± 1730 | 5720 ± 1280 |
| | SON | 6160 ± 1060 | 6320 ± 990 | 6500 ± 1060 | 6310 ± 1390 | 6550 ± 1120 | 6570 ± 990 | 5830 ± 1890 | 6590 ± 1620 | 6430 ± 1280 |
| Depth (m) | DJF | 200 ± 300 | 200 ± 300 | 300 ± 200 | 200 ± 200 | 200 ± 300 | 300 ± 300 | 200 ± 200 | 300 ± 300 | 200 ± 100 |
| | MAM | 200 ± 100 | 200 ± 200 | 200 ± 200 | 200 ± 200 | 200 ± 200 | 200 ± 200 | 200 ± 200 | 200 ± 100 | 200 ± 200 |
| | JJA | 200 ± 300 | 200 ± 200 | 200 ± 200 | 200 ± 300 | 200 ± 100 | 200 ± 200 | 200 ± 200 | 200 ± 300 | 200 ± 100 |
| | SON | 200 ± 200 | 200 ± 200 | 200 ± 200 | 200 ± 200 | 200 ± 200 | 200 ± 100 | 200 ± 200 | 200 ± 200 | 200 ± 200 |
| Strength (°C/100m) | DJF | 0.34 ± 0.3 | 0.35 ± 0.23 | 0.33 ± 0.23 | 0.32 ± 0.24 | 0.37 ± 0.32 | 0.43 ± 0.32 | 0.36 ± 0.26 | 0.34 ± 0.27 | 0.44 ± 0.26 |
| | MAM | 0.32 ± 0.19 | 0.34 ± 0.25 | 0.35 ± 0.25 | 0.32 ± 0.19 | 0.41 ± 0.29 | 0.38 ± 0.29 | 0.38 ± 0.28 | 0.43 ± 0.26 | 0.45 ± 0.32 |
| | JJA | 0.34 ± 0.23 | 0.36 ± 0.28 | 0.33 ± 0.22 | 0.34 ± 0.29 | 0.37 ± 0.21 | 0.39 ± 0.26 | 0.4 ± 0.34 | 0.43 ± 0.29 | 0.41 ± 0.25 |
| | SON | 0.43 ± 0.25 | 0.44 ± 0.41 | 0.44 ± 0.3 | 0.44 ± 0.28 | 0.46 ± 0.31 | 0.47 ± 0.31 | 0.4 ± 0.39 | 0.49 ± 0.31 | 0.5 ± 0.38 |
| Strength (°C/depth) | DJF | 0.95 ± 3.02 | 0.86 ± 2.6 | 0.9 ± 1.84 | 0.61 ± 1.06 | 1.08 ± 3.66 | 1.26 ± 2.89 | 0.76 ± 1.36 | 0.9 ± 1.94 | 0.67 ± 0.71 |
| | MAM | 0.49 ± 0.62 | 0.91 ± 3.37 | 0.74 ± 2.49 | 0.52 ± 0.86 | 0.85 ± 2.09 | 0.95 ± 2.94 | 0.93 ± 2.52 | 0.7 ± 0.88 | 0.94 ± 1.53 |
| | JJA | 0.99 ± 4.06 | 0.86 ± 2.65 | 0.57 ± 0.95 | 0.87 ± 5.48 | 0.52 ± 0.77 | 0.67 ± 1.47 | 0.69 ± 1.87 | 1.11 ± 3.58 | 0.63 ± 0.92 |
| | SON | 0.82 ± 1.99 | 0.97 ± 3.68 | 0.82 ± 1.37 | 0.95 ± 2.56 | 1.04 ± 2.91 | 0.79 ± 0.98 | 0.78 ± 1.42 | 0.98 ± 1.65 | 1.34 ± 5.57 |
| Number of inversions | DJF | 138 | 82 | 244 | 253 | 108 | 186 | 348 | 115 | 61 |
| | MAM | 231 | 124 | 192 | 259 | 195 | 179 | 169 | 113 | 77 |
| | JJA | 203 | 111 | 217 | 279 | 135 | 197 | 258 | 104 | 83 |
| | SON | 315 | 155 | 277 | 306 | 160 | 216 | 288 | 101 | 91 |



As with the low-level inversions, the seasonal variability in elevated inversion characteristics presented in Table 3, were not
representative of radiosonde measurements over Walvis Bay given in Table 4. Mean elevated inversions at Walvis Bay were
highest in the spring (5940 ± 1850 m) and summer (5990 ± 1420 m) and lowest in the autumn (5260 ± 1370 m) and winter
(5020 ± 2070 m). We did however see a similar high winter variability in base heights along the coast in both datasets.
**Table 4: Seasonal mean and standard deviation of the base heights (m amsl), depths (m), strengths (°C/100 m) and strength**
**throughout the depth (°C/depth) of elevated inversions (2.5 – 10 km) from radiosondes released at Walvis Bay airport.**

| | | 2.5 – 10 km |
|---|---|---|
| **Height (m agl)** | Su. | 5990 ± 1420 |
| | A. | 5260 ± 1370 |
| | W. | 5020 ± 2070 |
| | Sp. | 5940 ± 1850 |
| **Depth (m)** | Su. | 60 ± 40 |
| | A. | 70 ± 40 |
| | W. | 90 ± 50 |
| | Sp. | 70 ± 40 |
| **Strength (°C/100m)** | Su. | 0.01 ± 0 |
| | A. | 0.01 ± 0.01 |
| | W. | 0.01 ± 0 |
| | Sp. | 0.01 ± 0.01 |
| **Strength (°C/depth)** | Su. | 0.55 ± 0.56 |
| | A. | 0.44 ± 0.45 |
| | W. | 0.73 ± 0.78 |
| | Sp. | 0.52 ± 0.42 |
| **Number of inversions** | Su. | 44 |
| | A. | 65 |
| | W. | 82 |
| | Sp. | 52 |

The smallest diurnal variability in GPS-RO detected base heights was along the coast over the cold, upwelling waters, and
towards lower latitudes (Figure 15). Variability was greatest over the subcontinent. Across the entire region, the number of
inversions was lowest at noon and in the afternoon (Table S.3). There was also a slightly higher incidence of deeper inversions
during this time, although mean depths remained around 0.2 ± 0.3 km. Inversion strengths per 100m were higher in the morning
and at night (Table S.3 and Figure 16). Variability in inversion strength throughout the depth was greatest along the coast and
smallest over the ocean (Table S.3).

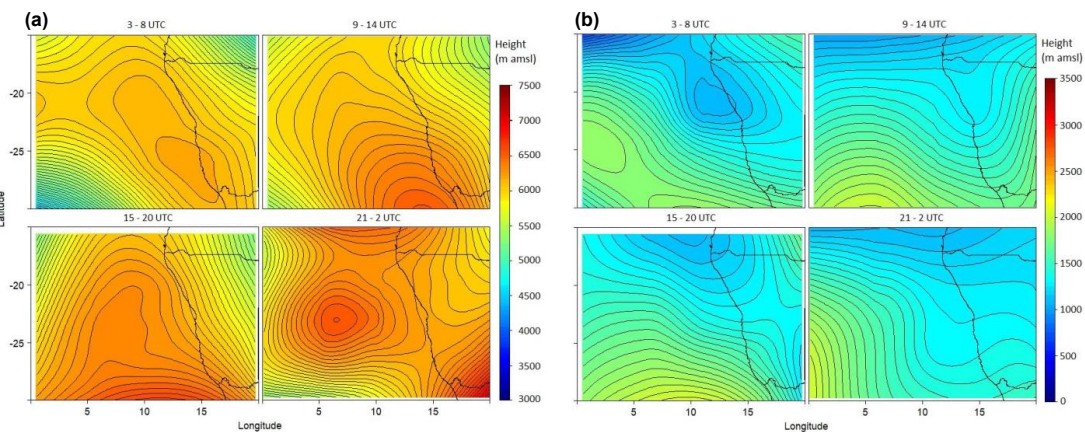

**Figure 15: The diurnal (a) mean and (b) standard deviation of elevated inversion base height over the area of interest. Polynomial smoothing was applied to the data.**




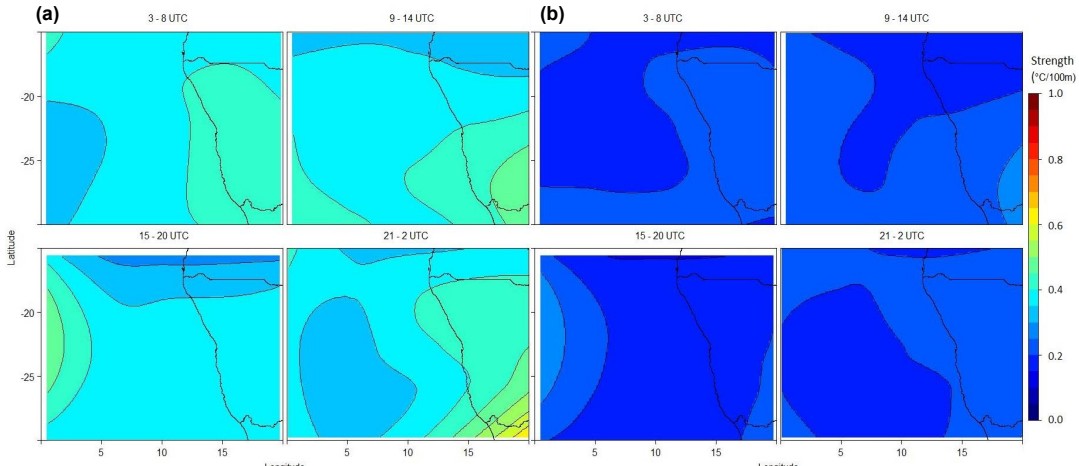

**Figure 16. The diurnal (a) mean and (b) standard deviation of elevated inversion strength over the area of interest. Polynomial smoothing was applied to the data.**

### 5.4. Co-occurring inversions

Multiple layers of temperature inversions in the same profile were frequent over the study region. Instances of these co-occurring elevated inversions between 0.5–10 km, indicated as a percentage of the total number of inversions measured, are summarised by time of day (Table 5) and month (Table 6). Annually, co-occurring inversions identified from GPS-RO profiles were most frequently measured along the coast (16.8%), then over the ocean (14.2%) and least frequently over the subcontinent (9.9%). Sixteen percent of inversions measured by radiosonde at Walvis Bay were co-occurring with another in the same profile. These frequencies were less than the one in five reported by Cosijn and Tyson (1996) for the region along the coast.

**Table 5: The percent of inversions co-occurring with another in the same profile as a function of the total, summarised by the time of day and location.**

|  | Morning | Noon | Afternoon | Night |
|---|---|---|---|---|
| **Ocean** | 14.6 | 14.9 | 13.6 | 13.8 |
| **Coast** | 18.0 | 14.8 | 14.8 | 17.4 |
| **Subcontinent** | 9.9 | 7.1 | 9.7 | 10.9 |

Over the subcontinent and along the coast, the instances of co-occurring inversions were at a minimum around noon and the afternoon, whereas the maximum instances were measured in the morning and at night. Over the ocean, there was little diurnal variability in the frequency of co-occurring inversions.

**Table 6: The percent of inversions co-occurring with another in the same profile as a function of the total, summarised for each month and over the ocean, coast and subcontinent.**

|  | Jan. | Feb. | Mar. | Apr. | May | Jun. | Jul. | Aug. | Sep. | Oct. | Nov. | Dec. |
|---|---|---|---|---|---|---|---|---|---|---|---|---|
| **Ocean** | 14.8 | 11.7 | 13.4 | 10.2 | 12.5 | 11.1 | 13.1 | 14.1 | 18.8 | 21.1 | 14.7 | 12.9 |
| **Coast** | 19.9 | 15.0 | 15.5 | 12.3 | 12.6 | 9.6 | 13.5 | 15.0 | 21.9 | 22.9 | 16.8 | 17.8 |
| **Walvis Bay** | 13.6 | 14.8 | 4.8 | 10.3 | 16.7 | 25.0 | 16.7 | 20.1 | 20.1 | 9.5 | 8.3 | 18.5 |
| **Subcontinent** | 9.4 | 7.5 | 8.2 | 7.9 | 10.1 | 6.6 | 7.9 | 9.6 | 15.7 | 14.5 | 9.6 | 7.7 |

Across all three regions from GPS-RO profiles, the maximum monthly frequency of co-occurring inversions was recorded in September and October (Table 6). Seasonal variability of frequencies was greatest along the coastal margin, with a maximum of 22.9% in October and a minimum of 9.6% in June. The inversions measured by radiosonde data at Walvis Bay also saw an increase in the frequency of co-occurring inversions between May and September. The general trend of the monthly variability across all four regions showed a sharp decline after the October maximum, except for high incidences of multiple inversions in the coastal margin in January. Monthly variability in co-occurring inversions was smallest over the ocean.





Co-occurring inversions were frequently observed between 0.6 and 1.5 km over the ocean. Along the coast, inversions were
co-occurring most between 0.6 and 1 km and then between 5.6 and 6.6 km. Over the subcontinent, the heights of frequently
co-occurring inversions were most variable, with frequent co-occurrences between 1.4 and 1.7 km, 5.4 to 5.7 km, 6 to 6.5 km
and 7.7 to 7.9 km.
**6.   Discussion**
**6.1.  Macroscale circulation**
Cosijn and Tyson (1996) have reported enhanced stability over the ocean and along the coastal margin, to the east and south
of the SEA anticyclone. The system exerts a year-round influence on circulation over the region and subsidence peaks in the
winter (Tyson and Preston-Whyte, 2014). Despite the influence of macroscale stability-inducing systems, wintertime
inversions were not necessarily stronger (Figure 11), deeper, nor more frequently observed in the GPS-RO data (Table 1) in
comparison with the rest of the year. We ascribe this to the disturbance of easterly waves (spring and summer) and a lesser
degree, the highly transient westerly waves (autumn and winter) (Preston-Whyte et al., 1977; Tyson and Preston-Whyte, 2014).
The seasonal variability in MG height is dampened at higher latitudes, where macroscale subsidence results in strong inversions
at BL top, and higher cloud fractions (Muhlbauer et al., 2014).
Over the region, quasi-stationary barotropic easterly low-pressures, which move in direct response to surface heating over the
semi-desert landscapes of southern Africa (Tyson and Preston-Whyte, 2014), fluctuate between 15° and 25°S (Tyson et al.,
1996). Towards the locus of these transient low-pressures, enhanced convective activity and the resultant instability disturb the
mean circulatory field between the surface and approximately 850 hPa, over central and northern Namibia (Garstang et al.,
1996; Tyson and Preston-Whyte, 2014). This explains the decreasing frequency of low-level inversions measured towards
lower latitudes and the low frequency of inversion formation over the subcontinent (Table 1). In the spring, and to a lesser
extent, summer, inversions still develop over the subcontinent under the influence of a band of suppressed convection that
forms between the approaching easterly lows and weakening influence of anticyclones (Cook et al., 2004; Preston-Whyte et
al., 1977).
Low base heights of elevated inversions over the subcontinent during the winter and autumn suggests a link to the migration
of the high-pressure belt, which was theorised by Cosijn and Tyson (1996). This includes the continental and SEA anticyclone
which adiabatically heat subsiding air and have been linked to the formation of inversions as high as 500 hPa (Preston-Whyte
et al., 1977; Cosijn and Tyson, 1996). Despite the increased subsidence and atmospheric stability induced by these high-
pressure conditions, the wintertime inversions were the weakest and springtime inversions strongest. We did however see an
increased formation of co-occurring inversions in the colder months as compared to the warmer months (Figure 13 and Table
3). The increasing frequency of co-occurring inversions between May and September coincides with the most intense
anticyclonic circulation over the subcontinent after which the variability decreased. The year-round cold SEA ocean and
subsidence under the SEA anticyclone, is responsible for higher frequencies of co-occurring inversions, than over the
subcontinent. The diurnal trends in stability over the subcontinent and coast were more pronounced than over the ocean, with
higher frequencies of co-occurring inversions at night and in the morning. Instances of co-occurring inversions decreased
towards the locus of the Sc cloud and lower latitudes, where circulation is influenced more by the easterly low pressures. The
zonal decrease in co-occurring inversions toward the south (Figure S.11), may be explained by the smaller influence of high-
pressure cells at lower latitudes and is also in good agreement with the decreasing concentrations of absorbing aerosol
concentrations (De Graaf et al., 2014; Zuidema et al., 2018).
Along the coast, the frequently occurring, morning and night-time inversions at low-levels may indicate the presence of the
weak counterpart of the low-level Benguela jet stream for which Nicholson (2010) reported a temperature lapse of 2°C between
850 and 700 hPa (approximately 1 450–3 000 m, ie. inversion strength of 0.13°C/100m). For the stronger counterpart,



Nicholson (2010) reported a temperature lapse of 7°C between 850 and 925 hPa (approximately 750–1 450 m, i.e. inversion
strength of 1°C/100 m). Of all those night-time inversions measured along the coastal margin, only 2.2% had strengths of at
least 1°C/100 m and may be linked to the formation of the stronger jet stream counterpart.

### 6.2. Differential heating

Differential heating across the cold ocean and coastal margin, and the mainly arid landscapes across the subcontinent, results
in the formation of steep temperature and pressure gradients across the region on diurnal and seasonal scales (Markowski and
Richardson, 2010; Tyson and Preston-Whyte, 2014). The resultant winds that form between the marine and continental
atmospheres are a significant modulator of the BLH along the coastal margin (Davis et al., 2020). Land breezes and berg winds
are known to enhance stability and typically form at night and in the winter near the surface and at the BL top (Ao et al., 2012;
Davis et al., 2020; Tlhalerwa et al., 2005). Evidence of this is seen in the high BLHs ($R_N$) and high variability in winter (Figure
4), which coincides with the increased frequency of occurrence in local easterly wind components below 500 m (Figure 5).
These breezes are sometimes sustained into the summer at the top of the BL (Lindesay et al., 1990), as seen sporadically in
Figure 5. Berg winds often co-occur with coastal lows that recirculate local air (Lengoasa et al., 1993; Tyson and Preston-
Whyte, 2014), which explains the high variability in BLH ($R_N$) seen in Figure 4, particularly in the winter.
Along with the effects on BLH, radiative heating and cooling and the resultant temperature gradients and varying specific heat
capacities of air masses across the region could explain several observed characteristics in the atmospheric vertical structure.
These include the increasing heights in MG and low-level inversion layers towards the subcontinent (Figure 7 and 9), higher
daytime MG height over the subcontinent, less pronounced temporal variability in MG and low-level inversion heights over
the ocean (Figure 8), a higher frequency of co-occurring night and morning inversions over the subcontinent and along the
coastal margin, and high diurnal variability in elevated inversion character, with higher morning and night-time strengths, and
daytime minimums in the number of inversions (Table S.4). Radiative cooling over cold upwelling waters along the coast may
also explain the relatively small diurnal variability in low-level inversion base height (Figure 11) and strength (Figure 12).
Cool-marine air masses are transported in sea breezes and plain-mountain winds. These form most commonly in the daytime
and during summer (Lindesay et al., 1990). During these times, we found the steepest zonal gradients in low-level inversion
strengths, with strengths decreasing towards the subcontinent (Figure 10 and Figure 12). Sea-breezes have been found to inhibit
the noontime convective development of the coastal BLH (Davis et al., 2020), but we could not detect this in our once daily,
morning measurements. This effect is also not apparent in the MG height (Figure 8), but it may be responsible for maintaining
low noontime low-level inversion heights (Figure 11).

### 6.3. Cloud fraction

The lifetime and fraction of low-cloud cover is a known modulator of the BLH over tropical coasts (Davis et al., 2020). In the
presence of clouds, some of the incoming solar radiation is intercepted at cloud top, limiting the convective heating at the
surface, and resulting in an overall lower BLH as compared to clear sky conditions (Davis et al., 2020), as seen in our low
BLHs ($R_N$) between October and March, when cloud fractions over the area are higher than for the rest of the year (Modern-
Era   Retrospective   analysis   for   Research   and   Applications,   Version   2   (MERRA-2),   available   at:
https://gmao.gsfc.nasa.gov/reanalysis/MERRA-2/; last accessed: 24/03/2021: Gelaro et al., 2017). The outflow of warm
continental air masses leads to the dissipation of cloud (Painemal et al., 2014). Our findings of higher means and high
variability in BLHs ($R_N$) under clear sky conditions between April and August, were in good agreement with the results of
Davis et al., (2020). Based on these findings, we can conclude that the variability in the BLH ($R_N$) over Walvis Bay is a function
of the origin and direction of airflow, and surface heating that, in combination with macroscale circulation, is also linked to
the frequency of occurrence and variability in cloud fractions.



Incident surfaces are subject to the maximum incoming solar flux at noon, particularly under clear sky conditions. As expected,
the effect of surface heating on diurnal variability of MG height, was more pronounced over the subcontinent than over the
ocean, with higher (lower) mean heights in the day (at night). Ho et al., (2015), reported a difference of 300 m between cloud
top and MG of refractivity, and a correlation of 0.62. We found some interesting links between the two on seasonal scales. The
meridional gradient in MG height is most (least) steep in the winter (summer) when cloud fractions over subcontinental
Namibia are at a minimum (maximum) and the fraction of disorganised MCC over the ocean are higher (lower) (Muhlbauer
et al., 2014; MERRA-2: Gelaro et al., 2017). In the autumn and winter, maximum MG heights coincide with the annual
maximums in disorganised MCC and open MCC, and minimums in closed MCC (Muhlbauer et al., 2014), in other words, low
cloud fractions. Likewise, the late winter, early spring minimum in MG height along the coast (and the lowest standard
deviations as compared to the rest of the year) (Figure S.4), coincides with higher fractions of closed MCC and lower fractions
in disorganised MCC. The smallest differences between the radiosonde and GPS-RO MG height were measured in March,
May, June, September, and October. The strong, deep, and low springtime low-level inversions over the ocean coincide with
the annual regional maximum in closed MCC (Muhlbauer et al., 2014), where Painemal et al., (2015) found the strongest
temperature inversion at the core of the Sc over the SEA. Unsurprisingly, considering the climatological variability of the SEA
Sc (Painemal et al., 2014), no relationship between this low-level cloud and elevated inversions could be found.
**6.4.  Biomass burning aerosols**
This study does not attempt to study, nor suggest any causal relationships between aerosols and atmospheric stability in the
region. We did, however, observe some interesting correlations between pollution plumes and inversion characteristics, as well
as the frequency of co-occurring inversions observed. Maximums in co-occurring inversions were measured in September and
October across all three regions, coinciding with the maximum in radiation-absorbing aerosols over Namibia (Eck et al., 2003).
There also exists the potential for cloud fractions in the region to be modified by aerosols above cloud (Costantino and Bréon,
2013; Dagan et al., 2016) and from within the BL (Diamond et al., 2018). Gupta et al., (2020) showed how cloud-top
entrainment of biomass burning aerosols (BBA) increased BLH for a September study over the region. Our findings of low
annual springtime mean heights in both BLH and MG height, however, did not reflect this. This may be because the entrainment
of BBA, and the variable effects on clouds, depends as much on the aerosol character and load, as location relative to cloud,
and the relative atmospheric conditions (Painemal et al., 2014; Zuidema et al., 2018).
The additional stratification, stabilisation and decoupling induced by multiple elevated inversions inhibit vertical motions and
result in the formation of both heavily polluted plumes and clean air slots, an observed but not yet clearly defined phenomena
over southern Africa and offshore (Hobbs, 2003). These clean air slots have been observed adjacent to plumes of pollution
(Hobbs, 2003) or cloud (Costantino and Bréon, 2010), and also above (Haywood et al., 2004) and below smoke layers (Wilcox,
2010) with varying differences in the thickness and location of those gaps (LeBlanc et al., 2020). The strong and high elevated
inversions, and strong, deep, and low springtime low-level inversions identified in this study (Figures 9 and 10 and Table 1)
coincide with the June to October maximum in the biomass burning over southern Africa (Chiloane et al., 2017; Swap et al.,
2003). Formenti et al. (2018) reported increased black carbon concentrations within the BL between April and September with
a maximum in June of 2017. These plumes have been observed over the south Atlantic as far Ascension Island between June
and October (Wu et al., 2020) and were specifically observed above clouds over the SEA in September and October (Zuidema
et al., 2018). Deaconu et al., (2019) showed a 1K increase in the temperature inversion at the top of the Sc deck offshore of
Angola when the aerosol optical thickness at 865 nm was greater than 0.04. The strong low-level springtime inversions over
the ocean and offshore of Angola may be an indication of this.



## 7. Conclusions

This study of 11 years of GPS-RO refractivity and temperature profiles, and three years of radiosonde data at Walvis Bay, provides an in-depth look at the spatial and temporal variability of atmospheric discontinuities over Namibia and the SEA Ocean. We discussed our findings in relation to the potential drivers of this variability, based on existing research specific to our region of interest. The direct comparison of temperature profiles revealed that GPS-RO temperatures generally underestimated those measured by radiosondes throughout the profile depth (up to 10 km). The mean differences in temperatures from the two datasets decreased with distance from the surface. This is attributed to errors in refractivity related to atmospheric moisture, which increases towards the surface. The identification of the point of minimum gradient in the refractivity profiles compared well with radiosonde data, but not for all cases. The reason for these discrepancies could not be determined, but was found to not be related to cloud top. The comparison also revealed that the MG was not consistent with BLH estimated using the bulk $R_N$ definition. The BLH estimated by the bulk $R_N$ definition was consistent with the height and variability in existing literature. This reaffirms the importance of identifying real structures consistent with BLH and not other structures that may share similar characteristics to the BL, such as sharp moisture and temperature gradients. The sensitivity of GPS-RO signal propagation to atmospheric moisture means that, despite identifying the large-scale variability, it would be useful for future investigations to report variability for cloudy and cloud-free conditions separately. Unlike previous studies, low-level inversions were observed over the subcontinent, although very rarely. We found links in the location and strength of the nighttime low-level inversion, to the previously reported strong counterpart of the Benguela jet stream. We also found similarities in the location of the mean strongest inversions over the ocean with the location of the inversion over the Sc. Finally, we found correlations between seasonal maximums in low-level inversion strength and maximum occurrence of co-occurring inversions, to the seasonal peak in biomass burning over the subcontinent (Eck et al., 2003; Swap et al., 2003). The effects of atmospheric circulation on a variety of spatial scales, as a result of latitudinal location, variability in topographic features, radiative interactions over different landscapes and low-level clouds in the region, were all found to correlate with- and contribute to the complex character of the BLH ($R_N$), MG height and low-level inversions. The elevated inversions were also influenced by subsidence under macroscale circulation systems, and diurnally varying surface radiative effects like the low-level inversions; however, no link between elevated inversions and low-level clouds were found. The high inversion strengths over the ocean and subcontinent in the spring coincide with the seasonal peak in BBA over the region. Considering that springtime inversions were stronger than inversions measured during the winter when subsidence under the high-pressure belt exerts the greatest influence on circulation over the region (evident in the significant decrease in base heights), it might be reasonable to suggest that the radiative characteristics of BBA plumes trapped within stratified layers, contribute to the high inversion strengths measured in the spring. Further research is required to investigate this hypothesis.

*Data availability.* COSMIC GPS-RO data and ECMWF data were obtained from the COSMIC Data Analysis and Archive Center (CDAAC), available at http://cosmic-io.cosmic.ucar.edu/cdaac/index.html. The openair package for R is found in Carslaw and Ropkins (2017). Radiosondes were collected and provided by the Namibia Meteorology Service. MODIS Atmosphere L2 Cloud Product (06_L2), http://dx.doi.org/10.5067/MODIS/MYD06_L2.061.

*Author contributions.* DK analysed the data with contributions by RB and SJP. DK wrote the paper with contributions from SJP, RB and PF. SD made the measurements and compiled the radiosonde data.

*Competing interests.* PF is guest editor for the ACP Special Issue "New observations and related modelling studies of the aerosol–cloud–climate system in the Southeast Atlantic and southern Africa regions". The remaining authors declare that they have no conflicts of interests.

*Acknowledgements.* We thank the Namibian Meteorology Service for funding the collection and archiving of the radiosonde data.



***Financial support.*** D. Klopper acknowledges the financial support of the Climatology Research Group of North-West
University and the travel scholarship of the French Embassy in South Africa. This work receives funding by the French Centre
National de la Recherche Scientifique (CNRS) and the South African National Research Foundation (NRF) through the
"Laboratoire International Associé Atmospheric Research in southern Africa and the Indian Ocean" (GDRI-ARSAIO) and the
Project International de Coopération Scientifique (PICS) "Long-term observations of aerosol properties in Southern Africa"
(contract n. 260888) as well as by the Partenariats Hubert Curien (PHC) PROTEA of the French Minister of Foreign Affairs
and International Development (contract numbers 33913SF and 38255ZE).

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
