# Peer review of "Atmospheric stratification over Namibia and the southeast Atlantic Ocean"

_Atmospheric Chemistry and Physics, 2021_

## Author Comment (AC1)

Atmos. Chem. Phys. Discuss.,
https://doi.org/10.5194/acp-2021-668-RC1
https://doi.org/10.5194/acp-2021-668-RC2

**Response to comments on "Atmospheric stratification over Namibia and the southeast Atlantic Ocean" by Danitza Klopper et al.**

The authors would like to thank the two anonymous referees for taking great efforts and time to read and provide very useful and constructive criticism on the manuscript. Both referees commented on the poor structure and flow in the paper (and the need for improvements), and the need to clearly outline the scientific contribution of this work. Both referees also agreed that if the comments would be fully addressed and the paper re-worked, it could make relevant contributions to our broader understanding of atmospheric stratification over the southeast Atlantic and Namibia. We are extremely grateful for the extra eyes and inputs and have made extensive efforts to address the comments from each referee in detail (starting with referee #1 followed by those of referee #2). Each comment is listed below along with our *accompanying responses* below that. Sections inserted into the paper are given in blue.

Comments from both authors motivated for;

- The restructuring of the paper for improved flow and ease of understanding.
- Clear communication of the scientific contribution.
- Improved connection between the satellite and the radiosonde portions of the analysis.
- Improved connection between the different BLH definitions just using the radiosonde data.
- Re-evaluation of the calculation of superrefracted signals, which was changed from initially being determined from a smoothed gradient over a sliding window, to rather identifying the individual points and removing those profiles.
- A change of the area plots (showing BLH and inversions across latitude and longitude) to zonally averaged plots where the regional variability across diurnal and seasonal scales is easier to identify and compare.
- The results and discussion sections were combined to present our findings alongside the relevant literature and in the context of the broader meteorological picture.

After incorporating the responses from both referees, the manuscript was certainly improved and we hope that it is now acceptable for publication in ACP.
* * *
**Referee #1: specific comments**

1. The paper begins with a discussion of the COSMIC satellite data and its processing algorithm. For those not familiar with the COSMIC satellite data, Section 3 is rather difficult to follow regarding which processing (Abel inversion algorithm, "*atmPrf*" dataset, ECMWF "*1-D var*" moisture correction?) is provided or what is additional processing/analysis done by the authors e.g. following the Shyam reference? It's not really clear in Section 3.1 which processing was done by the authors in the present work, and which was in an external dataset (as the references seem incomplete).
**AND** Sections 3 and 4 seem disjointed, as, for example, Section 3.1 talks about the data processing for COSMIC and Section 4.1 also talks about COSMIC data processing and definitions, and Sections 3.2 and 4.2 both discuss radiosondes.
**AND** Would it be better to combine the two COSMIC GPS-RO sections, and the two radiosonde processing sections into one Data/Methods section, followed by analysis of the results?

*Following the suggestions and in the interest of improved structure, the entirety of section 2 (Data and Methods) has been reworked to clarify the pre-processing already done on the dataset by ECMWF and that post-processing performed for the article (Section 2.2.), and definitions used in data analyses (Section 2.3).*

2. The paragraph starting on Line 134 seems to belong more in the background/introduction sections.

*You are correct and the literature from these sources was incorporated into the text in the introduction and results sections.*

3. Section 5.1.1: The conclusions section states that the GPS-RO method underestimated temperatures in the temperature profiles, but that doesn't seem to be strongly supported by Section 5.1.1 (an absolute error of -0.3+/- 1.3C seems fairly evenly distributed between positive vs negative differences) or Figure 2, which shows pretty good agreement between the two

methods at least as it is presented there (see below "other comment" about Fig 2 as well). I'd recommend finding a different visualization if the point of this figure is to say that one method systematically underestimates temperature.

*We added Table 1 to show a bit more detail about the over- and under-estimations between the pairs of co-located temperature profiles. We also showed the correlations between points in the co-located profiles and believe the conclusion is now better written to communicate our results.*

4. The authors mention it multiple times, but it's not clear how the issue of superrefractivity might be affecting the analysis. Line 100: "*even with the applied corrections, no reliable information about atmospheric structure can be collected below where the signal is super-refracted*": where is this point, typically, and how frequently do these conditions occur in the region of interest?

*We have improved the flow and motivation behind this sentence by adding, "For this reason, we don't consider information in N profiles beyond the point where the critical value of -157 N/km is exceeded (Sokolovskiy, 2003). A total of 1802 out of 32223 profiles contained superrefracted signals; 1619 out of 19937 profiles over the ocean, 198/6561 over the coastal margin, and 24/6525 over land. Superrefraction appeared most frequently at 0.6 km ± 0.2 km. In this study, 60% of the profiles were usable from 100 m agl, and 20% from 200 m."*

5. In Section 4 the authors mention three or four separate methods to calculate BLH only from the radiosondes, but the refractivity definition compared with the other three is never really explored. It's mentioned briefly on Line 494 that it isn't consistent with the RN definition, but by then we're already in the conclusions. To my mind this needs to be addressed far before that because the analysis of Section 5 uses both refractivity definitions and inversion definitions, so those need to be reconciled. What are the considerations of each calculation? What are we supposed to take away from these different definitions beyond "the BLH can vary rather widely based on what definition of BLH you use" (this issue of definition was also discussed somewhat in this same special issue by Ryoo et al, https://doi.org/10.5194/acp-2021-274).

*Thank you for this comment. The exploration of the BLH and different methods to calculate it has now been more clearly laid out in section 2.3. Data analysis > 2.3.1. Boundary layer height. In the results section, we also first introduced the comparison between the BLH identified from the primary height in break point (same definition) for the two datasets. Then we went on to show the results from the radiosondes (local scale) and those from the GPS-RO (regional scale) separately. We also updated the discussion on how these different methods for estimating BLH compare to one another.*

6. The value of the refractivity definition is understandable in that it allows a direct comparison to the satellite-based retrieval (see above), but Figure 3 shows it doesn't do a particularly good job in that respect, and Figure 4 shows that the other three definitions aren't consistent with one another either. So what's the use of any of this? And the somewhat arbitrary throwing out of 6 points in Fig 3 doesn't lend any confidence to the time series of these parameters in Fig 6 either. I'd first suggest a clear delineation of each BLH calculation description, either as a bulleted list or maybe even a table.

*Our radiosonde data (and the updated methods used there) show rather good agreement, but yes, it is true that the height in primary breakpoint, as identified in the literature (by authors who did global studies) did not agree well with the "traditional" methods for calculating BLH from radiosondes. We have included all the co-located pairs and instead just discussed different groupings, i.e. those that we more closely related in distance, time, or differences in estimated BLH.*

7. Also, come up with clear names/abbreviations for each of the BLH definitions (e.g., one of them is described as "the point where the virtual potential temperature (VPT) aloft is the same as at the surface," multiple times, when you could just call it BLH_VPT or something similar after Section 4). It's difficult to keep all of those straight. **AND** Line 141 says "the point of MG of refractivity (hereafter MG height)" but later in the text uses "MG height" and "the height of the MG of refractivity" (Fig 7) and "the height of MG N-refractivity profiles" (Line 269) etc… are these all the same thing?

*Clear, concise abbreviations for the different definitions of BLH have been defined as per the suggestion.*

8. And "low-level inversions" and "surface-based inversions" are the same? It's quite hard to follow.

*These are not the same, and we believe the reworking of the sections on BLH and temperature inversions now do a better job of clarifying this difference.*

9. Section 5.1.2: I see the value in comparing the refractivity BLH calculations for GPS-RO and radiosondes, but Figure 3 doesn't seem to support that these are comparable. In this section the authors eliminate several potential explanations for the poor agreement, but then exclude the worst-comparing points based on nothing other than they are the worst-comparing points. What's to say that the majority of points in Fig 6 don't show that same discrepancy, then? How can these really be compared?

*Thank you for this comment. The differences in BLH estimation as the primary height in break point of the refractivitiy profile are now shown in* **figure 2** *and discussed in* **3. Results and discussion > 3.1. Comparisons between COSMIC GPS-RO and radiosonde data around Walvis Bay > 3.1.2. Comparison of boundary layer heights**. *The investigation shows that, indeed, the differences between BLH estimations in the two datasets showed, "The majority of the co-located pairs had an absolute difference within 300 m, not entirely unreliable considering the 100 m resolution of the GPS-RO profiles. These biases are in the order of the annual average biases between 300 and 500 m reported for the southeast Atlantic (Ao et al., 2012; Basha et al., 2018)." Furthermore, in our investigation of BLH and low-level temperature inversion identified from GPS-RO profiles, we found, "Over the entire region, low-level inversions were on average higher than primary $h_{BP}$ by $280 \pm 400$ m (88% of the time) and lower than primary $h_{BP}$ by $570 \pm 480$ m in the same profile."*

*To address the issue of potential explanations for the differences, we looked at the correlations between differences in BLH estimated by the two datasets, and other factors like; cloud fraction at the point where the profile was taken, difference in time of measurement (up to 6 hours), and distance between measurements. We updated the text to read, "For the eleven co-located pairs with differences in BLH estimation smaller than 300 m, the correlation of differences in $h_{BP}$ estimation to month increased to 0.46, with a moderate relationship, where the differences in BLH estimations were smallest between June and October. For the eight remaining pairs with differences greater than 300 m, there's an even higher correlation of 0.73 between month and biases in height, but in these cases the biggest differences in BLH estimation were found in July and August. The high variability in wintertime biases between the measurements might be explained by the increase in differential heating during these months, and fewer cloud fractions resulting in a less steep gradient and therefore less prominent BLH in the GPS-RO refractivity profile. The exploration into these relationships confirms that the differences in BLH estimation over the region is not solely a function of distance between these co-located pairs, nor the sharp moisture gradients like those between the mixed layer and overlying Sc cloud."*

10. Throughout Section 5.2 and 5.3, the analysis jumps back and forth between GPSRO and radiosonde analysis, under the headings of "*Spatial and temporal variability*" although really only the GPS-RO data can give spatial variability here, right? Given the results of the earlier sections, it seems to me the takeaway is that they aren't really interchangeable, although the structure of Tables 1/2 and Tables 3/4 make it rather difficult to compare the results from the two methods. **AND** Confusingly, Sections 5.2.3 and 5.3.1 are both titled "*temperature inversions*" but refer to either low-level or mid-level temperature                                                                                                                   inversions. **AND** It's not clear how spatial plots of low-level temperature inversions are derived from only the GPS-RO data, given the superrefractivity questions above and the clear altitudinal limitations of these data as shown in Fig 2, especially relative to the radiosonde-based inversion height in Fig 4.

*We believe the restructured format removes any confusion in this regard.*

11. I'd also move Fig 6 up to significantly earlier in the paper, e.g. just after Fig 3, as they are showing similar things and the context for how GPS-RO and radiosondes compare with one another is a necessary prerequisite before talking about the spatial and temporal patterns in their results.

*As suggested, the comparison was presented first, followed by BLH estimated from radiosonde and then GPS-RO data.*

12. Finally, it's difficult to see how these observations (which I think are worth describing if the above issues can be addressed) fit into the broader meteorological picture, which I think is what the authors are trying to do in Section 6. These connections are tenuous at best. Most of the spatial maps presented (Figs 7-16) primarily show seasonally-averaged values and then standard deviations (or try to; the stdev figures are extremely hard to interpret, see additional comments below), so it's not clear to me how this relates to transient meteorological events e.g. as discussed in Sections 6.1 and 6.2. Section 6.3 discusses cloud fraction but the spatial analysis will rely on the GPS-RO profiles which are less reliable in cloudy conditions, is that right? How is this addressed? The authors mention MERRA-2 (and also mention MODIS in the "data availability" section but apparently nowhere else in the paper?), I can't help but think that a comparison of the profiles here with a large-scale reanalysis or model that gives atmospheric motion (MERRA-2 or perhaps ERA5 which performs better in the region; see Ryoo et al., 2021 https://doi.org/10.5194/acp-2021-274 or Pistone et al., 2021 https://doi.org/10.5194/acp-21-9643-2021) is necessary if the goal of this work is to contextualize these boundary layer height variations within the larger context of the regional atmospheric circulation.

*Thank you for this comment. The purpose of this paper is to gain a better understanding of the variability (monthly and seasonal) in thermodynamic structure of the regional atmosphere and relate these findings to what other investigators have found. The text in the introduction reads as follows, "In this paper, we aim to present an 11-year (2007 – 2017) climatology of the characteristics and variability of the boundary layer height and atmospheric inversions over Namibia and the SEA below 10 km above ground level (agl). The motivation for this is to understand the features of these layers within the lower troposphere, which influences the*

*dilution and transport of atmospheric gases and aerosols.*" *We also attempted to identify potential mechanisms of formation that describe what we see in terms of the temporally averaged variability but this was not the main aim.*

*In regards to the presence of cloud, GPS-RO can penetrate cloud, and can still collect reliable information within cloud (with slightly increased uncertainties), but considering that all points below superrefraction have been removed, these remaining data (irrespective of the presence of cloud) are included in the analyses.*

*We have not done additional modelling, but do believe that it would be beneficial to do these types of analyses - we have included this as a recommendation for future research in the conclusions.*

*The high standard deviations are indicative of skewed data, which yes, might be better described using other statistical summaries, but not in all cases. In consideration thereof, we included frequency distribution plots (Fig. S.2. and S.5.) which helped to illustrate this. Furthermore, the results and discussion sections were combined to present our findings alongside the relevant literature and in the context of the broader meteorological picture.*

*Furthermore, we checked the data for super refracted points and updated the text at* **2.2. Data collection and processing > 2.2.1. Global Positioning System - Radio Occultation data** *as follows, "It is important to note that even with the applied corrections, information about the atmospheric structure is not reliable below the point where the refractivity signal is superrefracted (Sokolovskiy, 2003). For this reason, information in N profiles beyond the point where the critical value of -157 N/km is exceeded are not included in the dataset (Sokolovskiy, 2003). A total of 1802 out of 32223 profiles contained superrefracted signals; 1619 out of 19937 profiles over the ocean, 198/6561 over the coastal margin, and 24/6525 over land. Superrefraction appeared most frequently at 0.6 km ± 0.2 km. In this study, 60% of the profiles were usable from 100 m agl, and 20% from 200 m. The frequency of these valid profiles is summarised in Figure S.1, which shows an uneven spatial and temporal distribution across the three regions identified in Figure 1."*

*Regarding the reanalysis data and placing of results in the context of the macroscale circulation, we plotted geopotential heights at different levels in the atmosphere (in figure A below). These plots did not add anything new to the analyses apart from what the existing literature states, and will therefore not be included in our manuscript.*

[Figure]

**Figure A:** The geopotential mean height of the atmosphere between 2007 and 2017 over the greater region of interest (ERA-5:Copernicus Climate Change Service (C3S), 2017)

**Additional comments**

13. It's not very clear to me what you're trying to convey with Figure 2. If the focus is on the BLH difference between the two datasets, then why show the full altitude scale up to 10.5km? It's very difficult to see what the differences are between ~2-5km. On the other hand, if the point is to show that the lapse rate is generally in agreement, I don't think you need 36 panels to do that (also, I'd suggest making the lower line thicker, it's really difficult to see the underlying blue line there). These aren't every single coincident profile, correct? How many are in the circle shown in Figure 1? Lines 112-113 indicated 4007 within the coastal region, were only 32 comparable?

*We changed the criteria for co-located profiles, and ultimately identified 19 valid profiles within 100 km and 6 hours of one another. This is not too small a number of pairs, considering that we have 442 profiles collected in the noon and afternoon along the coastal margin between 20 and 25 for the total eleven year period. We decreased our initial co-location criteria from 200 to 100 km since the point of the exercise is to see how well the profiles compare since we could encounter high variability in the thermodynamic profile across greater distances, increasing uncertainties in our findings.*

14. Also, why so many more valid retrievals the ocean? Based on just the surface area shown in Fig 1, I'd expect maybe 2-3 times as many profiles over the ocean versus the continent… is there a further difference in what makes valid retrievals for land vs ocean beyond just the atmospheric moisture? I'd mention that.

*We couldn't find an explanation for why there are more retrievals over the ocean other than the relatively larger size of the sampled area. Due to this, it was also difficult to say whether there were relatively more inversions forming over the ocean than the rest of the region.*

15. How many retrievals are going into Fig 4? Are there systematic differences in May vs June re: the 10am vs 9am launch time? Or could the sharp increase in the RN BLH range between those two be due to diurnal BLH development, or fewer radiosondes being launched in May 2015 vs April or June?

*All figures have been updated with the number of retrievals. Regarding the low BLH in May, we updated the text at **3.2. Boundary layer heights > 3.2.1. Seasonal variability** to include the following, "The minimum BLHs in May, may in part be due to fewer measurements, or the increase in easterly wind components between 2.5 and 10 km (Fig. 4). At mid-levels, the inflow of warm continental air masses over the cool, moist Walvis Bay, forms a cap enhancing low level stability and increasing cloud fractions (Ryoo et al. 2021), translating in a lower BLH."*

16. Figure 5: I'd recommend a different color scheme especially for panel A; wind direction of 0 = 360 degrees, so having one be red and the other being blue is difficult to interpret e.g. northerly from easterly (also I'd recommend adding to the caption 0=east for ease of reading, assuming that's the convention being used here).

*Thank you, in consideration of this, we have replotted the figure with a looping colour scale and indications of cardinal directions on the key.*

17. The 2x8 figures are overall very difficult to interpret; it's not at all clear what is the main message of each of these very similar figures. Beyond that, the color scale on Figs 7b, 8b, 9b, 11b, 14b, and 16b makes it extremely difficult to interpret, beyond "*they're all small*". If that's the message, you can lose all these panels altogether. If it's not, then a different scale should be used to show the variations between different panels. **AND** are the black contours the same parameters? At what interval are those lines? And how much data is included in these figures (how many overpasses; is this also limited to mid-morning or is this all times of day; are the retrievals regularly distributed in time and space or are there particularly retrieval-rich times or overpasses which could bias the results preferentially towards a certain time or condition)?

*The black contours (or colour intervals) are indeed the same parameters and divide the heights into 100 m segments. Considering the difficulty of interpretation of these plots, however, after consultation with co-authors we have decided to use zonally averaged line graphs depicting the mean and standard deviations to present the results. The number of data points included in the figures are however now included in the figure captions for clarity. Please refer to **Fig S.1**. in the supplementary material for the sampling frequency by time of month and location.*

18. Relatedly, it's not clear what the authors are intending to convey with the standard deviations throughout the paper, especially when the ranges are much larger than the mean values themselves. For example, how can you have an inversion depth of 200m +/- 300m or a temperature inversion strength of 0.55 +/- 0.56 C/depth [depth = 60 +/- 40 meters?]? Isn't that saying a nonnegligible fraction of the data would have zero to- imaginary temperature inversions? I think another metric of variability might be more instructive, either in terms of percentiles or just showing the frequency distributions of select parameters.

*The high standard deviations are indicative of skewed data, which yes, might be better described using other statistical summaries, but not in all cases. In consideration of this, we included supplementary figures S.2. and S.5., showing frequency distribution of boundary layer and inversion heights summarised by region, season and time of day.*

19. Also, Figures 6-11 (except 10) show heights above mean sea level in a region where ground level is ~1-2km (e.g. Fig 1)? I'm not sure amsl is the most instructive height metric here. If "mean MG heights were consistently higher over land" (Line 265), were they relatively higher compared to magl?

*This is true, and all presentations of results were converted to height above ground level so that all measurements are comparable from the surface of the earth.*

20. Figure 8: Why show the seasonal variability of the refractivity BLH in Fig 6, but show the diurnal variability this way? For all the discussion about diurnal variability (e.g. also Section 6.3), I think this would be better served as a time series. As with other figures, I'd like to know how many retrievals go into each of these.

*The number of retrievals that go into plots have been provided in each panel. Although the figures were updated, we still have the BLH at Wlavis Bay (one location point) presented as a time series, and those layers identified in the GPS data (across the region) are presented as zonal averages.*

21. Section 6.4: if it is decided to keep in the larger discussion, is there any indication regarding whether the presence of aerosols would affect the validity of the GPS-RO profiles, as humidity/clouds do?

*Considering that GPS profiles are most affected by moisture content (with less bias in dry atmospheres), and superrefracted profiles were excluded from the analyses, we don't see any reason why aerosols in elevated layers should affect GPS retrievals.*

22. Line 38: Is "subcontinent" a common term to refer to this part of Africa? It seems to be more just the southern continent-proper.

*Yes it is, but for clarification, we'll stick to using "southern Africa", "land", or "Namibia" throughout the text.*

23. Lines 120-125: this is a bit confusing. Was the primary set of radiosondes always at 10am local, with an additional set between 10 and 11? **AND** Also, why say you're converting time to UTC, and then describe the dataset in local time?

*We updated the text for clarity, as follows, "These radiosondes were generally launched at 10:00 local time, with some launches at 09:00. We also include 232 radiosondes from 2014 and 2016, released between 10:00 and 11:00 local time, for additional comparisons of estimated boundary layer heights and temperature profiles with GPS-RO data. Radiosonde timestamps, reported in local Namibian time, are converted to UTC for comparison with GPS-RO profiles and to study diurnal trends across a standardised measure of time. The conversion is done as follows; UTC +1 for measurements between the first Sunday in April to the first Sunday in September, otherwise UTC +2".*

*We describe the dataset in local time ranges that were set based on the local variability in solar radiation. Our divisions are as follows: morning (3 to 8 UTC), noon (9 to 14 UTC), afternoon (15 to 20 UTC), and night (21 to 2 UTC), where e.g. morning measurements translate to 4/5 am to 9/10 am local time (+1 or +2 depending on the seasons), noon measurements translate to 11/12 to 15/16 pm (when we have the greatest amount of radiative heating locally), etc.*

24. Line 199: isn't very low vapor pressure = very dry conditions, not a moist atmosphere?

*You are correct, thank you and my apologies for this oversight.*

**References**

Ao, C.O., Waliser, D.E., Chan, S.K., Li, J.L., Tian, B., Xie, F. and Mannucci, A.J.: Planetary boundary layer heights from GPS radio occultation refractivity and humidity profiles, J. Geophys. Res., 117(16), 1–18, doi:10.1029/2012JD017598, 2012.

Basha, G., Kishore, P., Venkat, R.M., Ravindra, B.S., Velicogna, I., Jiang, J.H., Ao, C.O.: Global climatology of planetary boundary layer top obtained from multi-satellite GPS RO observations, Clim. Dyn., 52(3), pp. 2385-2398, 10.1007/s00382-018-4269-1, 2018.

Copernicus Climate Change Service (C3S) (2017): ERA5: Fifth generation of ECMWF atmospheric reanalyses of the global climate . Copernicus Climate Change Service Climate Data Store (CDS), date of access. https://cds.climate.copernicus.eu/cdsapp#!/home

---

## Author Comment (AC2)

Atmos. Chem. Phys. Discuss.,
https://doi.org/10.5194/acp-2021-668-RC1
https://doi.org/10.5194/acp-2021-668-RC2

**Response to comments on "Atmospheric stratification over Namibia and the southeast Atlantic Ocean" by Danitza Klopper et al.**

The authors would like to thank the two anonymous referees for taking great efforts and time to read and provide very useful and constructive criticism on the manuscript. Both referees commented on the poor structure and flow in the paper (and the need for improvements), and the need to clearly outline the scientific contribution of this work. Both referees also agreed that if the comments would be fully addressed and the paper re-worked, it could make relevant contributions to our broader understanding of atmospheric stratification over the southeast Atlantic and Namibia. We are extremely grateful for the extra eyes and inputs and have made extensive efforts to address the comments from each referee in detail (starting with referee #1 followed by those of referee #2). Each comment is listed below along with our *accompanying responses* below that. Sections inserted into the paper are given in blue.

Comments from both authors motivated for;

- The restructuring of the paper for improved flow and ease of understanding.
- Clear communication of the scientific contribution.
- Improved connection between the satellite and the radiosonde portions of the analysis.
- Improved connection between the different BLH definitions just using the radiosonde data.
- Re-evaluation of the calculation of superrefracted signals, which was changed from initially being determined from a smoothed gradient over a sliding window, to rather identifying the individual points and removing those profiles.
- A change of the area plots (showing BLH and inversions across latitude and longitude) to zonally averaged plots where the regional variability across diurnal and seasonal scales is easier to identify and compare.
- The results and discussion sections were combined to present our findings alongside the relevant literature and in the context of the broader meteorological picture.

After incorporating the responses from both referees, the manuscript was certainly improved and we hope that it is now acceptable for publication in ACP.
* * *
**Referee #2: specific comments**

1. The results section presents values that sometimes have larger uncertainties than the average values themselves. Such large relative uncertainties render the results useless for practical purposes. For example, the authors find an error of -0.30 ± 1.30°C for temperature from GPS-RO when compared to the radiosonde below 7 km amsl. The same is true for cases above 7km or for the scatter plot in Fig. 3 (790 ± 90 m).

*The high standard deviations are indicative of skewed data, which yes, might be better described using other statistical summaries, but not in all cases. In consideration of this, we included additional frequency distributions (Fig. S.2. and S.5.) which helped to illustrate this, and we updated the text at 3.2. Boundary layer heights > 3.2.1. Seasonal variability to include, "...and the frequency distributions are given in Figure S.2. In general, there is a gradual increase in primary hBP over the ocean away from the coastal margin, and an abrupt increase in primary hBP from the coastal margin over land. Figure S.2. illustrates the high variability of BLH over land, and the preferential formation of lower BLH over the ocean and coastal margin." and at 3.3. Temperature inversions > 3.3.1. Seasonal frequency, height, strength, and depth, we added "Cold marine surfaces further enhance regional atmospheric stability, where seasonal variability is minimal and inversions frequently form at low-levels along the coastal margin between 500 and 750 m (Fig. S.5.), and over the ocean between 500 and 1250 m (Fig. S.5.), decreasing in height towards lower latitudes (Fig. 7 and Tab. S.1.)."*

2. Almost all the figures lack detailed information that can let the reader better understand exactly what they are meant to convey. In addition, the authors showed seasonally averaged figures of standard deviations. Some of these figures are not discussed in the text. All Figures and Tables (main text and supplementary): Please all figure captions should be full and complete -- Meaning that it should include all the information used to make the plot (time range, date name, and so on), regardless of

whether that information has been stated in the text or not. Also, all acronyms/abbreviations should be defined, irrespective of whether it has been used elsewhere or not.

*Thank you for this. All figure and table captions were updated accordingly.*

3. Finally, the authors used several different methods to calculate the inversion. However, while discussing them in the results section, the authors sometimes did not clearly specify which of the three methods they referred to.

*Only one method was used to calculate the inversions, and three were used to calculate the BLH, for which clear, concise abbreviations have been defined and used in the text to remove any uncertainty in this regard.*

**Additional comments**

4. Line 8-9: Why? Why is there "a limited understanding of the spatial and temporal variability in vertically stratified atmospheric layers over Namibia and the southeast Atlantic"? Please be more specific.

*For clarity, this was updated in the introduction as follows, "Above the boundary layer, the spatial and temporal features of temperature inversions over the region have not been extensively studied using high spatial and temporal resolution data, despite their importance for the elevated transport of aerosols over the west coast of Namibia."*

5. Line 18-21: This sentence says that the two profiles have a "good agreement" and then says that one profile underestimates the other. It is either one or the other. I suggest the authors remove the "good agreement part", and rewrite the entire sentence for better clarity.

*The entire section was re-written and placed in the context of other author's findings. We also explored the reasons for these differences in context.*

6. Line       14:       minimum       gradient       or       minimum       vertical       gradient?
   **AND** Line 24: What does it mean to "found correlations in the character"? That statement needs to be clarified.

*The abstract was re-written for clarity. The only other mention of "minimum gradient" was changed to "minimum vertical gradient".*

7. Line 33-57: It is difficult for me to understand the point of this introduction. I will suggest that the authors rewrite it, paying close attention to telling the readers exactly why they should care about this study.

*The introduction was re-written as suggested with a greater focus on the importance of the research and finished off with the intended contribution of this study.*

8. Line 78: What a priori information? This place needs appropriate references.

*To clarify, we updated the section on 2.1. Region of investigation to include, "This region along the coastal desert represents a transition zone between the cold Benguela current and arid Namibia (Preston-Whyte, Diab and Tyson, 1977; Cosijn and Tyson; 1996; Garstang, et al. 1996; Tyson and D`Abreton, 1998; Ao et al., 2012)" along with the relevant references.*

9. Line 83: "several times a day"? What time? You could provide a temporal interval.

*The supplementary materials include Figure S.1: Frequency of COSMIC GPS-RO measurements by month and year made over the ocean, coastal margin and land. Additionally, the number of measurements made across 4 four 6-hour intervals over the region are given in Tables S.4, S.5, S.6.*

10. Line 90: "The Abel inversion algorithm was applied…." By who? Additionally, the whole sentence should be rewritten for clarity.

*Following the suggestions and in the interest of improved structure, the entirety of section 2 (Data and Methods) has been reworked to clarify the pre-processing already done on the dataset by ECMWF and that post-processing performed for the article (Section 2.2.), and definitions used in data analyses (Section 2.3).*

11. Line 81 - 86: These lines mentioned "data" several times, without clearly specifying what data. Is this what the instrument measure? What exactly is it? What are the "raw data" separate from the "atmPrf " dataset that the authors mentioned?

*The Data collection and processing (now section 2.2) has been updated to better explain this, thank you.*

12. Section 4: There are several places where the words like "define" or "definition" were used to signify the calculation of the boundary layer height. For example, in line 149, the authors stated that "….was performed based on four different definitions of BLH….". I supposed the authors meant the four different methods used to calculate BLH. Would you please rewrite this section to reflect the right language that can better improve the clarity?

*These sections were re-worked as suggested and the use of "definition" was changed to "method" where appropriate.*

13. Line 187 & Fig. 2: The text mentioned that the temperature profiles from GPS-RO are taken for the same day as the radiosonde. Is this an average over the entire day or the one with measurement time closest to the soundings? Please clarify.

*The text was updated to specify "...co-located GPS profiles measured within 2 hours and 200 km of the radiosonde release from Walvis Bay…"*

14. Line 200: Delete "where". **AND** Line 221: Change "high cloud fractions" to "high fractions" **AND** Line 135-136: "… were on average 100 m higher in the autumn and 100 m lower in the spring" than what?

*These sections of text were significantly reworked and language edits were also made throughout considering reviewer comments and to ensure clarity in the text.*

15. Line 220-224: The authors should examine (and possibly include in the supplementary document) the cloud distribution/variability for the periods that are compared.

*Thank you for this suggestion. The supplementary materials were updated to include Figure S.4. Seasonally averaged cloud fractions for daytime measurements made between 2007 and 2017 (1° x 1° grid).*

16. Line 224-227: Given the large discrepancies and the fact that the authors have no explanation for the 6 data points, I don't believe they can make this conclusion based on the exclusion of those "bad" 6 points

*Thank you for this comment, you are correct. We have included all the co-located pairs and instead just discussed different groupings, i.e. those that we more closely related in distance or time.*

17. Line 238: "equivalent" or similar? Also, the BLH values are not similar, the BLH for VPT is about twice that of surface-based inversion. However, the monthly variation or monthly consistency is similar. The authors should rephrase this sentence.

*The entire section was re-written and hopes to provide clarity on these issues.*

18. Line 252-254: Comparing all data and not a subset of the data sounds like a more "sensible comparison" to me.

*Thank you for this suggestion. After consideration, all available months of data (2014-2016) were included on the condition that there were more than 10 measurements available for that month.*

19. Line 254: Do you mean the monthly variability? Fig. 6 shows monthly variability, not inter-annual variability. I also notice this in another part of the text. Please change all of them accordingly.

*You are correct, and the text was updated throughout for clarity and consistency.*

20. Line 254-255: how different would this estimate/assessment be if all data were included?

*Considering that the methods for calculating BLH were re-evaluated and a more regionally relevant set of methods were chosen, all analyses were re-done and results presented in such a way to consider the comments above.*

21. Line 266: Given that you talked about the difference between land and ocean, I wonder if this statement refers to zonal gradient and not "meridional" gradient.

*Meridional gradient is across lines of longitude, i.e. from ocean to land, so meridional is the correct use here.*

22. Line 491-493: This attribution was stated as speculation within the text. To make this type of conclusion requires more than speculation. I will suggest the authors better clarify their statement or remove it altogether.

*The bigger differences between temperature near the surface has been shown (by other authors) to be due to the higher humidity, as in our case as well.*

23. Line 505-508: Again, I don't see where this analysis was laid out in the result section of this paper. If the authors are making speculation, they either have to back this by previous studies that have made this conclusion or clearly state that they are speculating.

*We trust the improved organisation and discussion addressed this issue, thank you.*

**References**

Ao, C.O., Waliser, D.E., Chan, S.K., Li, J.L., Tian, B., Xie, F. and Mannucci, A.J.: Planetary boundary layer heights from GPS radio occultation refractivity and humidity profiles, J. Geophys. Res., 117(16), 1–18, doi:10.1029/2012JD017598, 2012.